# Fluidics system for resolving concentration-dependent effects of dissolved gases on tissue metabolism

Varun Kamat[1†], Brian M Robbings[1,2†], Seung-Ryoung Jung[1], John Kelly[3], James B Hurley[2], Kenneth P Bube[4], Ian R Sweet[1]*

[1]University of Washington Medicine Diabetes Institute, University of Washington, Seattle, United States; [2]Department of Biochemistry, University of Washington, Seattle, United States; [3]VICI Metronics, Seattle, United States; [4]Department of Mathematics, University of Washington, Seattle, United States

**Abstract** Oxygen ($O_2$) and other dissolved gases such as the gasotransmitters $H_2S$, CO, and NO affect cell metabolism and function. To evaluate effects of dissolved gases on processes in tissue, we developed a fluidics system that controls dissolved gases while simultaneously measuring parameters of electron transport, metabolism, and secretory function. We use pancreatic islets, retina, and liver from rodents to highlight its ability to assess effects of $O_2$ and $H_2S$. Protocols aimed at emulating hypoxia–reperfusion conditions resolved a previously unrecognized transient spike in $O_2$ consumption rate (OCR) following replenishment of $O_2$, and tissue-specific recovery of OCR following hypoxia. The system revealed both inhibitory and stimulatory effects of $H_2S$ on insulin secretion rate from isolated islets. The unique ability of this new system to quantify metabolic state and cell function in response to precise changes in dissolved gases provides a powerful platform for cell physiologists to study a wide range of disease states.

## Editor's evaluation

This paper presents a flow method for measuring the effects of dissolved gases on tissues while having control over tissue concentration. Working with gases can be challenging. The improvements reported here incorporate technology that allows for metabolic characterization of mammalian tissues while precisely controlling the concentration of abundant gases (e.g., oxygen), as well as trace gases (e.g., hydrogen sulfide).

## Introduction

### A critical need for instrumentation to study the effect of dissolved gases

Oxygen ($O_2$) is a fundamental determinant of cell survival and function in mammalian tissues. In most cells, the majority of ATP is generated by oxidative phosphorylation, driven by a series of redox reactions in which $O_2$ is the ultimate electron acceptor. Hypoxia is linked to many diseases including stroke, cancer, and diabetic complications. In addition to $O_2$, trace gases produced by cells ($H_2S$, NO, and CO) act as signals to regulate cellular and mitochondrial function (*Prabhakar and Semenza, 2012*). Ischemia–reperfusion injury is a condition common to many disease states and it is thought that during reoxygenation a burst of reactive $O_2$ species (ROS) occurs that can damage proteins, lipids, and nucleic acids (*Chouchani et al., 2016*; *Zweier et al., 1987*). Yet despite the scientific and clinical importance of dissolved gases, quantitative methods to measure the real-time effects of dissolved

**\*For correspondence:** isweet@uw.edu

[†]These authors contributed equally to this work

gases on intact tissue are not available. Investigators who have studied trace gases and who are characterizing drugs to attain the same benefits (*Chen et al., 2020*) almost exclusively use aqueous based surrogates/donors of gas. The equivalence of these drugs to the gases they are supposed to mimic has not been tested (*Chen et al., 2020*). Some investigators have bubbled gas directly into media, but this precludes adding essential protein to the media due to foaming. In addition, the study of dissolved gases is hampered by the volatility of dissolved gases under conditions where the headspace is not supplied with equilibrium levels of the gas (*DeLeon et al., 2012*). Thus, there is a strong need to develop technology that enables the study of both abundant and trace dissolved gases. The system we describe here does these analyses both quantitatively and reproducibly.

## A flow culture/assessment system that exposes tissues to precise levels and durations of dissolved gases

We developed a flow culture system according to three fundamental and essential specifications needed to assess effect of gases on tissue: (1) maintain tissue function and viability under continuous flow culture conditions; (2) continuously monitor parameters that reflect intracellular changes in metabolism in real time; (3) precisely control the aqueous and gas-phase composition of the media bathing the tissue. Although there are many systems readily available that have some components needed to investigate effects of dissolved gas, none incorporate all three. Commercially available hypoxia chambers (for instance Baker Ruskinn Cell Culture Workstations) control steady-state levels of dissolved gas and have been effectively and most commonly used for hypoxia studies (*Jaakkola et al., 2001*; *Epstein et al., 2001*). Microfluidics systems have been developed for establishing cell and tissue models where the three-dimensional structure and cell-to-cell interactions of native tissue can be recreated, which can be used under steady-state gas compositions (*Ren et al., 2013*; *Manafi et al., 2021*; *Kimura et al., 2018*). In addition, some investigators have used the Oroboros machine to characterize the effects of $O_2$ levels on metabolic processes (*Stepanova et al., 2019*). However, none of these methods are designed for implementing, and assessing real-time effects of, rapid changes in dissolved gas concentrations. The Seahorse flux analyzer measures OCR and extracellular acidification rate (mostly from glycolysis and the TCA cycle) on cell monolayers (*Wu et al., 2007*) and has been extensively utilized across many fields. However, it is not designed to maintain tissue in physiological buffers or to control dissolved gas levels.

In previous reports, we described an earlier version of our flow culture system and demonstrated its ability to maintain a range of tissues (including islets, retina, liver, and brain) over hours and days (*Neal et al., 2016*; *Neal et al., 2015*; *Sweet et al., 2009*; *Gilbert et al., 2008*; *Weydt et al., 2006*; *Bisbach et al., 2020*) while continuously assessing metabolic and functional effects of test compounds. This report highlights the incorporation of technology to precisely control both abundant gases (such as $O_2$, $CO_2$, and $N_2$), by using a countercurrent flow device that promotes equilibration between inflow media and premixed gas, and also trace gases, by novel application of permeation tubes. Permeation tubes are commonly used devices for calibration of safety equipment that detect toxic gases such as CO and $H_2S$. They consist of liquified gas housed under pressure in a metal jacket, where the gas continuously leaks through a membrane at a steady and accurately calibrated rate. We present in detail the components and operation of the system in the Methods section, and then illustrate the utility of the system to measure the effects of hypoxia followed by reoxygenation on two tissues, pancreatic islets and retina. We also demonstrate how this instrument can be used to quantify the effects of $H_2S$ on islet function and liver energetics.

## Assessment of $O_2$-sensitive processes: OCR, reductive state of cytochromes, and rate of lactate and pyruvate production

The most direct endpoints with which to assess the acute effects of $O_2$ are components of the electron transport chain (ETC), first and foremost OCR. However, measurements of OCR alone cannot identify the mechanisms that are mediating changes in OCR. Under physiological conditions, OCR can increase in response to changes in substrate supply (increased supply of electrons generated from metabolism) and/or demand (as stimulated by ADP *Chance and Williams, 1956b*; *Wilson et al., 1979*; *Figure 1*). Metabolism of fuels is reflected by a proportional change in cytochrome reduction and OCR: as the number of electrons bound to cytochromes increase, OCR increases by mass action (a 'push' system). In contrast, increased ATP usage by energy-utilizing cell functions (importantly ion

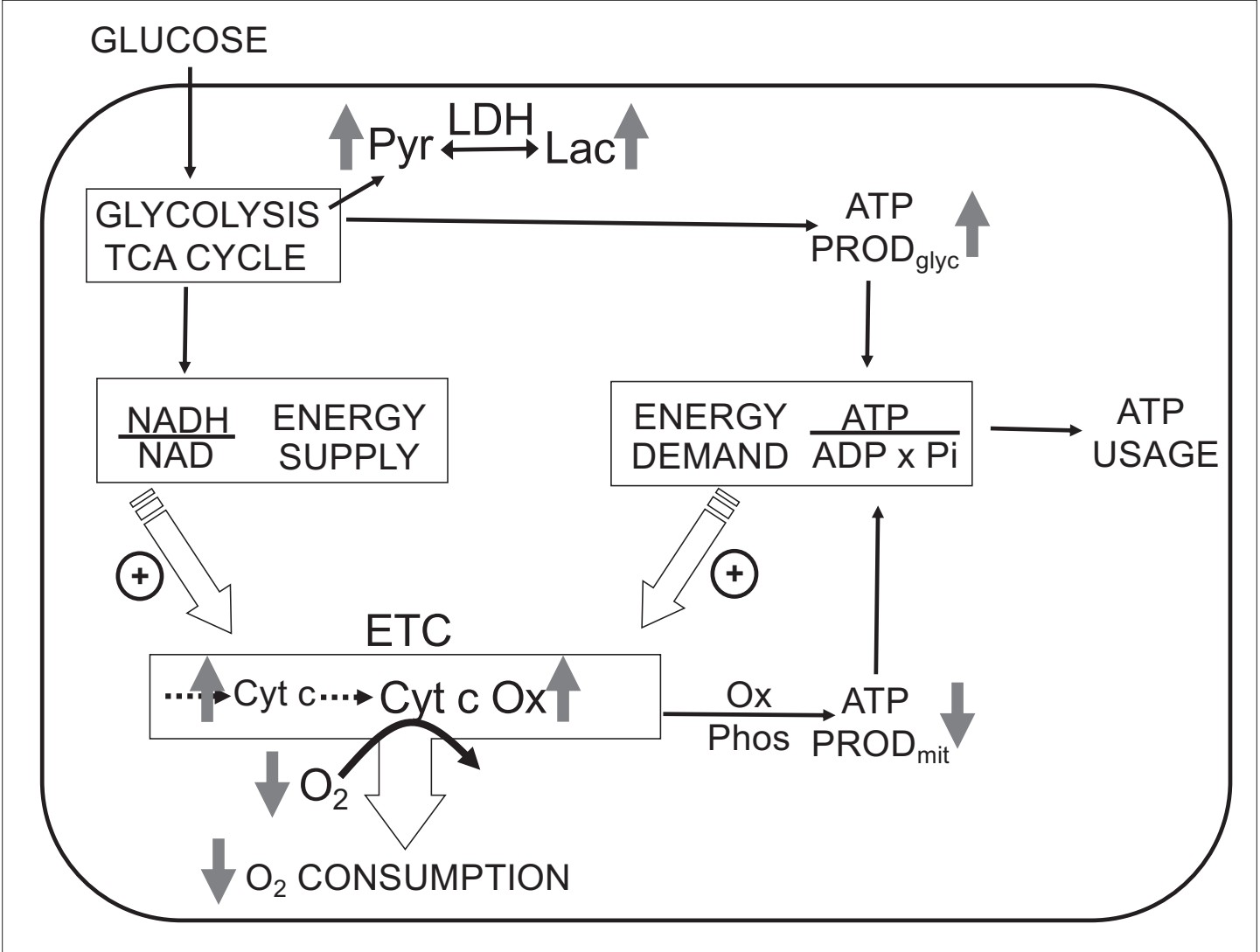

**Figure 1.** Direct control of $O_2$ consumption rate (OCR): energy supply vs. energy demand vs. $O_2$. *Schematic depicting three mechanisms mediating OCR*. (1) The supply of reduced electrons generated by metabolism of fuel. (2) The usage of ATP by energy-utilizing cellular function yielding ADP (a major regulator of OCR). (3) The concentration of dissolved $O_2$. The concomitant measurement of reduced cytochromes and $O_2$ allows for the distinction between the three mechanisms mediating observed changes in OCR. The vertical arrows depict the changes that are acutely affected by changes in $O_2$. Low $O_2$ leads to increased reductive state of cytochrome c and cytochrome c oxidase, decreased OCR. In some tissue types, the decrease in ATP production by oxidative phosphorylation is compensated by increased ATP generation from glycolysis (the Pasteur effect).

flux and biosynthesis) with a corresponding increase in ADP leads to increased OCR, but without increased cytochrome c reduction (a 'pull' system) (*Brown, 1992*; *Sweet and Gilbert, 2006*). In this way, OCR changes mediated by substrate supply vs. ATP usage can be distinguished, and these fingerprints are informative in understanding mechanisms mediating changes in the ETC induced by $H_2S$ and $O_2$.

In addition to regulating ETC activity, $O_2$ can also influence glycolysis (known as the Pasteur effect *Krebs, 1972*; *Figure 1*). Lactate and pyruvate accumulation and release are determined by relative rates of glycolysis, lactate dehydrogenase (LDH), pyruvate dehydrogenase, mitochondrial and plasma membrane transporters, and the redox state of the cytosol. Due to the equilibrium status of LDH, [lactate]/[pyruvate] ratio is proportional to the cytosolic redox state ([NADH]/[NAD]) (*Williamson et al., 1967*), so by measuring both lactate and pyruvate, changes in the rate of lactate production rate due to alterations in glycolytic flux vs. cytosolic redox state can be distinguished. The concomitant measurement of $O_2$, OCR, cytochrome c, cytochrome c oxidase, lactate, and pyruvate production

provides a comprehensive dataset to assess the multitude of biochemical and functional effects of $O_2$ and other gases.

## Assessment of H₂S effects on insulin secretion rate (ISR) from isolated pancreatic islets

Like $O_2$, $H_2S$ also interacts directly with the ETC, where it can be both stimulatory and inhibitory. $H_2S$ can inhibit cytochrome c oxidase (*Khan et al., 1990*) and it also can donate electrons to cytochrome c (*Vitvitsky et al., 2018*). However, with respect to pancreatic islets, all previous reports have described only inhibition of ISR (*Ali et al., 2007*; *Niki and Kaneko, 2006*; *Wu et al., 2009*; *Yang et al., 2005*; *Bełtowski et al., 2018*; *Tang et al., 2013*). Based on the ability of $H_2S$ to both increase and decrease ETC activity, we chose to demonstrate the technical caliber of our system by testing the hypothesis that $H_2S$ would both stimulate and inhibit ISR depending on its concentration. Flow culture systems are well suited to measure changes in ISR in response to changes in perifusate composition (*Lacy et al., 1972*). Secretogogs that affect ISR by isolated islets including glucose, arginine, amino and fatty acids, acetylcholine, GLP-1 as well as sulfonylureas and GLP-1 analogs, also manifest their effects in vivo. Therefore, isolated islets are a validated and highly relevant model with which to test our flow culture system. To evaluate the commonly asserted assumption that donors of $H_2S$ yield the same effects as direct exposure to $H_2S$, we also compared the effects of NaHS, a commonly used donor of $H_2S$ to direct exposure to dissolved $H_2S$.

## Results

## Measurement of OCR, reduced cytochrome c, and ISR by pancreatic islets in the face of changing inflow O₂

Ischemia–reperfusion is a stress to tissues that occurs under a range of pathophysiologic conditions, and it is recognized that damage from hypoxia can occur both from the period of decreased energy production and at the time when $O_2$ is replenished. Accordingly, we evaluated the ability of our system to measure the recovery of metabolism and function following a period of decreased $O_2$ levels. Isolated rat islets were placed into the perifusion chamber and perifused for 90 min with Krebs–Ringer bicarbonate buffer containing 3 mM glucose and equilibrated with 21% $O_2$/5% $CO_2$/balance $N_2$. Changes in OCR, cytochrome c reduction state, and ISR were measured in response to increased glucose (20 mM), decreased $O_2$ (by switching to a gas tank supplying the gas equilibration system that contained 3% for 2 hr), and the return of $O_2$ to 21% (*Figure 2*). To measure OCR, both the inflow and outflow $O_2$ concentrations were measured (*Figure 2A*), and the data were then processed by convolution techniques described in the Methods section. After transformation of the inflow $O_2$ using *Equation 4* and the transfer function generated with data obtained in the presence of potassium cyanide (KCN) to account for the delay and dispersion of the perifusion chamber, OCR was calculated from *Equation 5*. As expected, OCR increased with increased glucose concentration, and decreased with lower $O_2$ tension (*Figure 2B*). Notably, there was a transient spike of OCR when $O_2$ in the media was restored. OCR then approached a steady state that was about 55% of the prehypoxic level of OCR. One limitation of the system arises when defining the $O_2$ levels that tissue is exposed to, when in fact there is a gradient from the inflow to the outflow. This uncertainty can be minimized by increasing flow rate thereby decreasing the difference between inflow and outflow concentrations; however, as the difference gets smaller, the resolution of the method decreases.

The reduced state of cytochrome c was concomitantly measured with OCR (*Figure 2B*). Previous reports described an equilibrium with respect to the flow of electrons between NADH and cytochrome c (*Wilson et al., 1974a*; *Wilson et al., 1974b*), and their reductive state represents a balance between the supply of electrons generated by metabolism of fuels and use of electrons to drive proton translocation and ATP production. Consistent with these scenarios, glucose provided more reducing power to drive cytochrome c to its reduced state (in parallel with OCR), whereas hypoxia favored the reduced state of cytochrome c by slowing its oxidation. Following the return to 21% $O_2$, cytochrome c reduction reached a steady state of 42% of the prehypoxic levels, consistent with incomplete recovery of OCR. A particularly powerful feature of the systems approach is realized when tissue function can be measured concomitantly with measures of ETC. Fractions were collected during the protocol that

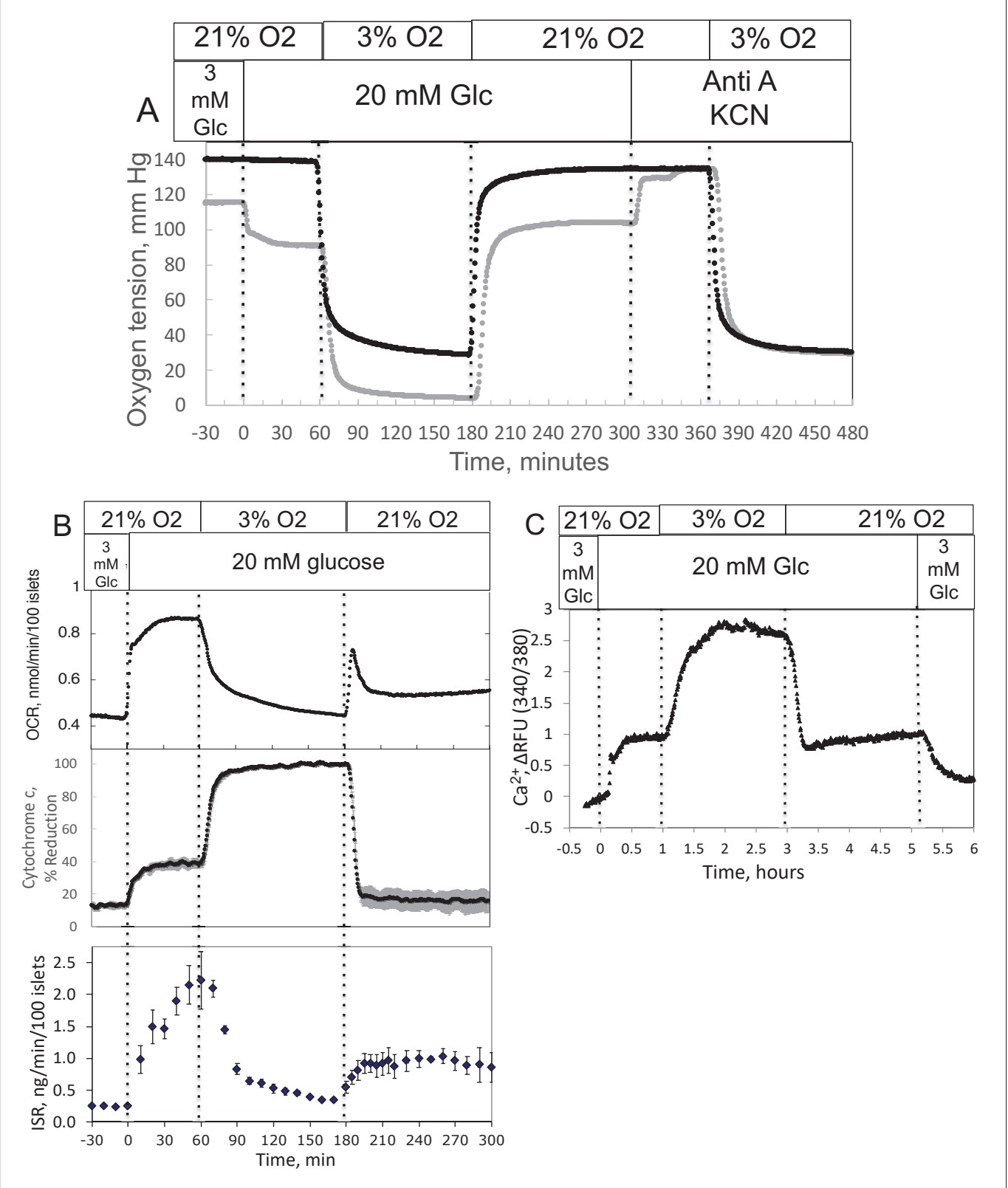

**Figure 2.** Effect of glucose and low $O_2$ on transient and steady-state electron transport chain (ETC), and insulin secretion rate (ISR) and intracellular $Ca^{2+}$ in pancreatic islets. Islets (900/channel) were handpicked with a P200 pipet and after mixing with Cytodex beads (1 µl/10 islets) loaded into the perifusion chamber, and the flow rate was set to 150 µl/min with Krebs–Ringer bicarbonate buffer containing 3 mM glucose for 90 min. At time = 0, glucose concentration was raised to 20 mM for 45 min; subsequently, $O_2$ was decreased to 3% for 2 hr, and then returned to 21%. (**A**) The protocol

*Figure 2 continued on next page*

*Figure 2 continued*

generated inflow and outflow $O_2$ profiles such as shown. Following the completion of the protocol, 12 µg/ml antimycin A (aA) was added for 25 min, and then 3 mM potassium cyanide (KCN), and the hypoxia protocol was repeated while islet respiration was suppressed in order to characterize delay and dispersion due to the separation in space of inflow and outflow sensors. (**B**) Calculated values of $O_2$ consumption rate (OCR) representative data from an *n* of 3 (average recovery = 0.55 ± 0.07), reduced cytochrome c (*n* = 2), and ISR (*n* = 2) were plotted as described in the Methods section. (**C**) In a separate illustrative experiment, intracellular $Ca^{2+}$ in islets was imaged and quantified using the same protocol except glucose was also decreased back to 3 mM glucose at the end of the experiment. Raw data can be found in a Source Data file named '*Figure 2—source data 1*'.

The online version of this article includes the following source data for figure 2:

**Source data 1.** Effect of glucose and oxygen on islet function, metabolism and signaling.

were later assayed for insulin (*Figure 2B*). Stimulation of ISR by glucose in the presence of 21% $O_2$ was suppressed in low $O_2$, and ISR recovered to 44% of its original level of stimulation after reoxygenation.

## Measurement of calcium ($Ca^{2+}$) in response to decreased $O_2$ by islets

As fluorescence imaging is a powerful and widely used modality to assess many intracellular signals including but not limited to $Ca^{2+}$ (*Rountree et al., 2014*), NADH (*Bennett et al., 1996*; *Pedersen et al., 2013*), mitochondrial membrane potential (*Pertusa et al., 2002*), ATP (*Hodson et al., 2014*), and ROS (*Neal et al., 2016*) in islets, we demonstrated the control of dissolved gas for this modality. We measured the effect of glucose and hypoxia on islet intracellular $Ca^{2+}$ with a protocol similar to the one we used for OCR except that glucose was lowered back to 3 mM at the end of the experiment (*Figure 2C*). As expected, intraislet $Ca^{2+}$ increased in response to the increase in glucose. Subsequently, in response to a decrease in $O_2$, $Ca^{2+}$ fluorescence rose 3-fold, presumably reflecting a loss of energy-dependent pumping of $Ca^{2+}$ out of the cells. In contrast to OCR and ISR parameters, $Ca^{2+}$ recovered fully to prehypoxic levels when $O_2$ was returned to normal levels.

## Measurement of lactate/pyruvate production and release by perifused INS-1 832/13 cells in response to changes in $O_2$

To track shifts between glycolytic and mitochondrial energy generation in real time, fractions collected from the outflow were assayed for lactate and pyruvate. We predicted that extracellular ratios of these two analytes reflect intracellular regulation of these two compounds. To evaluate this, we measured the response to inhibitors of LDH and mitochondrial transport of pyruvate in INS-1 832/13 cells (henceforth referred to as INS-1 cells). INS-1 cells were used instead of islets because most of the pyruvate made in islets is transported into mitochondria (*MacDonald, 1990*) since they do not have significant capacity for plasma membrane transport of lactate or pyruvate (*Ishihara and Wollheim, 2000*). Oxamate, an inhibitor of LDH, rapidly and completely suppressed lactate release from cells (*Figure 3A*), showing the tight relation between production of lactate from LDH and appearance of lactate in the outflow. Glucose-stimulated OCR obtained with INS-1 cells was similar to the values obtained previously (*Jesinkey et al., 2019*) and about half of what was measured using a Seahorse with metabolite-rich media (*Dover et al., 2018*). The rate of INS-1 OCR is about 1/4th of islet OCR on a per cell basis, probably reflecting loss of lactate in INS-1 cells that does not occur in islets, in addition to lower rates of synthesis and secretion of insulin. Somewhat surprisingly, pyruvate did not increase. However, OCR increased suggesting that the decrease in flux from pyruvate to lactate was counterbalanced by an increase in flux of pyruvate into the mitochondria. Blocking transport of pyruvate into mitochondria with [zaprinast a blocker of the mitochondrial pyruvate carrier (MPC; *Du et al., 2013*)] led to a rapid increase in pyruvate release from cells (*Figure 3B*) and diminished OCR. Why lactate production decreased is not clear from the data but could be explained by a decrease in the cytosolic redox state (NADH/NAD) by the increased activity of the malate/aspartate shuttle. Thus, the rapidity of the changes in lactate and pyruvate production in response to changes in LDH and MPC indicates that membrane transport is fast, and at least in this cell line, extracellular lactate and pyruvate will reflect changes in intracellular events controlling lactate and pyruvate with a time delay of no more than a few minutes. To ensure that this is the case, these experiments would have to be done in whatever tissue is being investigated.

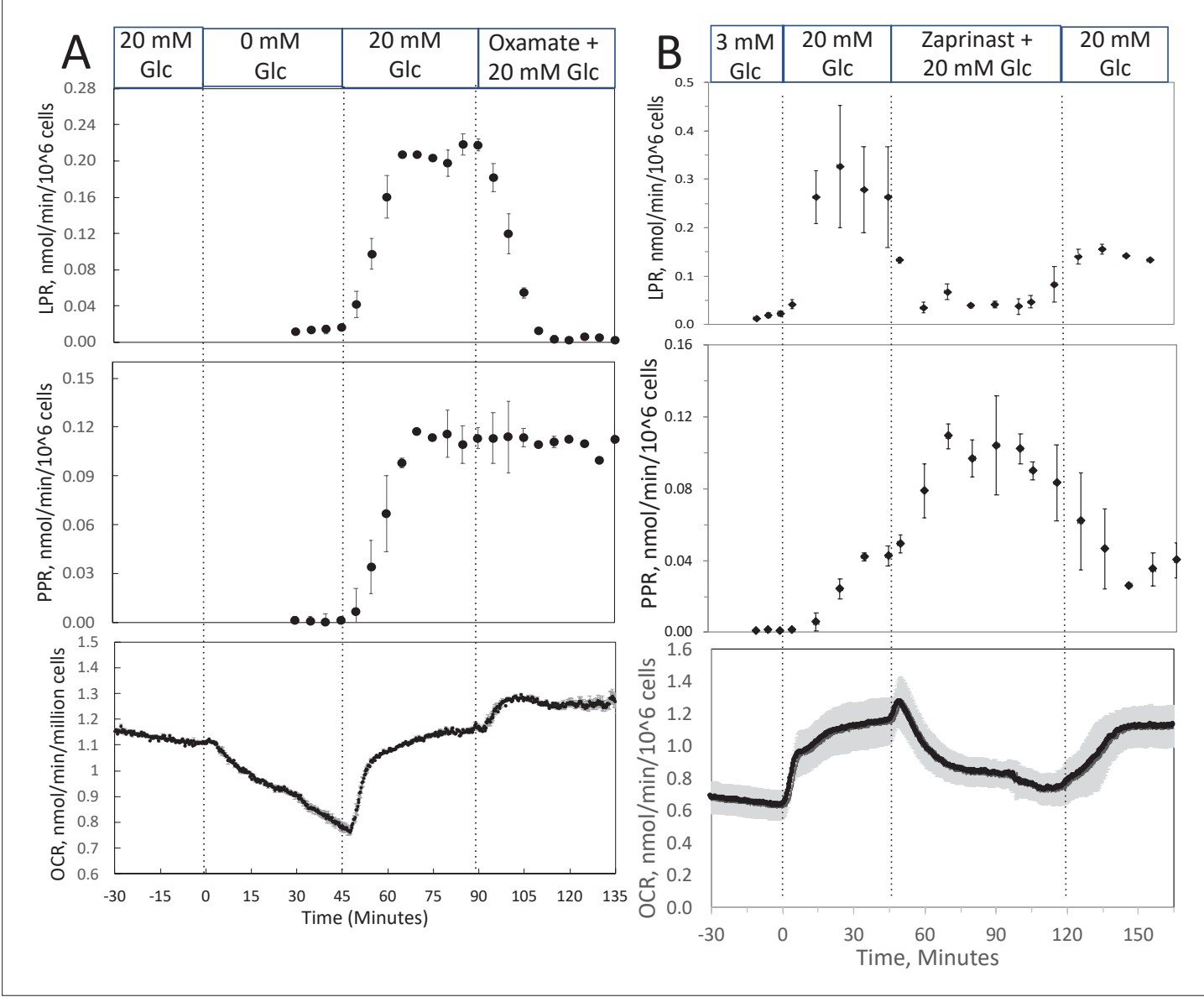

**Figure 3.** Effect of glucose and inhibitors of lactate dehydrogenase (LDH) and mitochondrial pyruvate carrier (MPC) on $O_2$ consumption rate (OCR), lactate, and pyruvate production rate by INS-1 832/13 cells. (A) The effects of glucose and then oxamate (50 mM, an inhibitor of LDH) on OCR, lactate production rate, and pyruvate production rate at the times indicated in the figure were measured. (B) The effects of glucose and then zaprinast (200 µM, an inhibitor of MPC) on OCR, lactate production rate, and pyruvate production rate. Data for both plots are the average ± standard error (SE), $n = 2$. Raw data can be found in a Source Data file named '*Figure 3—source data 1*'.

The online version of this article includes the following source data for figure 3:

**Source data 1.** Effect of low oxygen on metabolism in isolated retina.

## Measurement of OCR, cytochrome c, lactate, and pyruvate by perifused retina before and after a period of hypoxia

In order to compare results with islets to those obtained by a tissue that is less sensitive to hypoxia, experiments were carried out on isolated retina, a tissue that normally resides at low $O_2$ (*Bisbach et al., 2020*). Similar to the measurement in islets, the inflow and outflow $O_2$ were measured (*Figure 4A*), and OCR was calculated after convolution of the inflow data. OCR decreased at low $O_2$, and then manifested a transient spike in response to reoxygenation (*Figure 4B*). However, in contrast to islets, OCR by retina then approached a much higher recovery (a steady state of 83% of the prehypoxic rate). Reduced cytochrome c increased to maximal levels during 1% $O_2$ and stayed at this level throughout

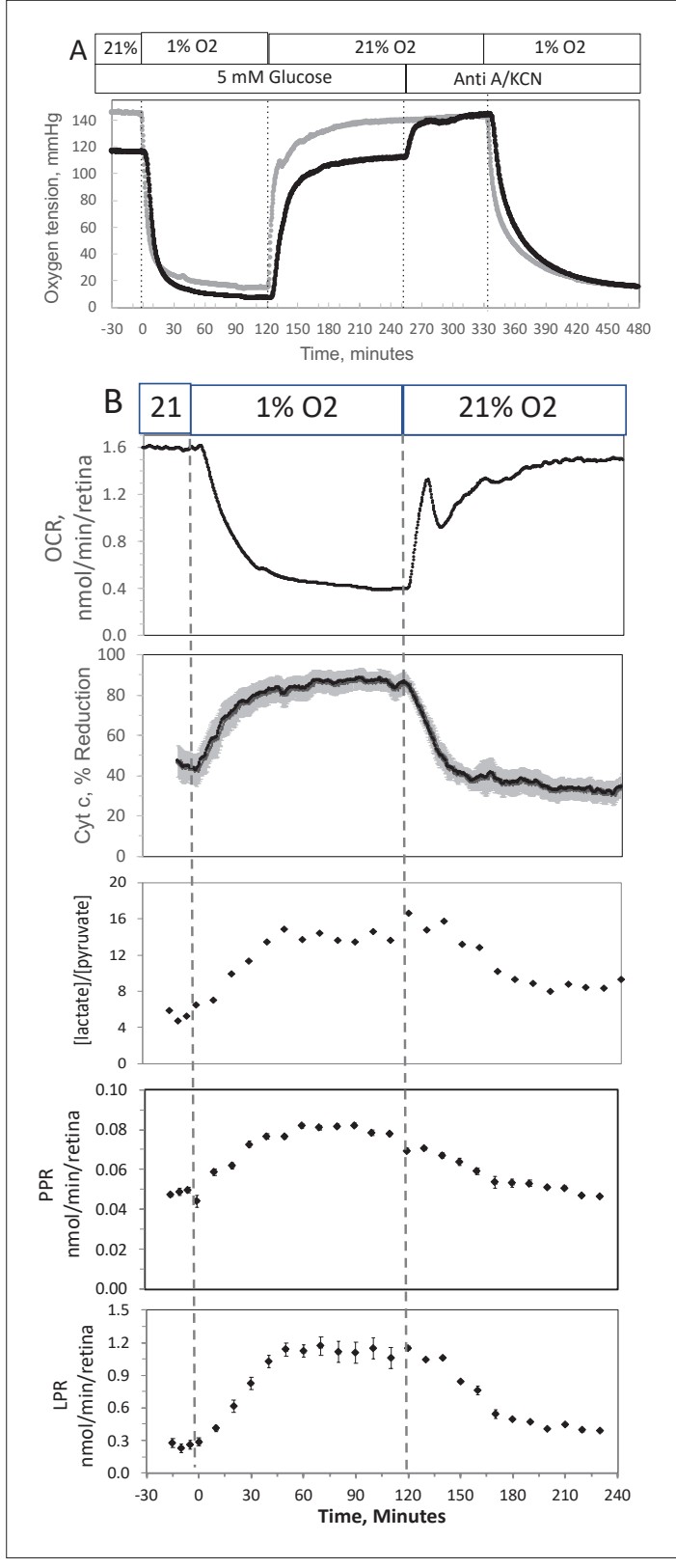

**Figure 4.** Effect of low $O_2$ on transient electron transport chain (ETC), and lactate/pyruvate in isolated retina. (A) Four retinas (16 pieces) per chamber were loaded into the flow chamber, where each group of 4 retinal pieces were separated by a 3 μl layer of Cytodex beads. The tissue was sandwiched on the top and bottom by 50 μl of Cytopore beads and both layers were held in place with a porous frit (Interstate Specialty Products, Suton, MA, Cat

*Figure 4 continued on next page*

*Figure 4 continued*

no. POR 4894, cut to 4.2 mm diameter and 0.25 in long). Ninety minutes after loading the retina into the system (flow rate = 130 µl/min), the $O_2$ tank was switched one containing 1% $O_2$ for 2 hr, and subsequently returned to 21%. (**A**) The protocol generated inflow and outflow $O_2$ profiles such as shown. Following the completion of the protocol, 12 µg/ml antimycin A (Anti A) was added for 20 min, and then 3 mM potassium cyanide (KCN), and the hypoxia protocol was repeated while retinal respiration was suppressed in order to characterize delay and dispersion due to the separation in space of inflow and outflow sensors. (**B**) Measurements of OCR representative data from an *n* of 6 (average recovery = 0.83 ± 0.03), reduced cytochrome c (*n* = 6), lactate and pyruvate production rates (*n* = 2), and [lactate]/[pyruvate] are shown. Raw data can be found in a Source Data file named '***Figure 4—source data 1***'.

The online version of this article includes the following source data for figure 4:

**Source data 1.** Effect of low oxygen on metabolism in isolated retina.

---

the 2 hr of low $O_2$ (***Figure 4B***). Consistent with OCR data, upon return of $O_2$ to 21% cytochrome c reduction recovered to 79% of prehypoxic levels. Comparing posthypoxia levels of OCR in retina vs. islets (***Figure 5***), islets did not recover from hypoxia as well as retina suggesting that the approach can be used to assess sensitivity to the stress of ischemia-reperfusion conditions. Note that due to the delay in time it took for the inflow perfusate to reach a new equilibrium, the inflow $O_2$ did not reach equilibrium levels with the $O_2$ from the supply gas tank. The levels of $O_2$ in the outflow are dependent upon the inflow $O_2$, the flow rate, and the OCR of the tissue in the chamber. In order to match the

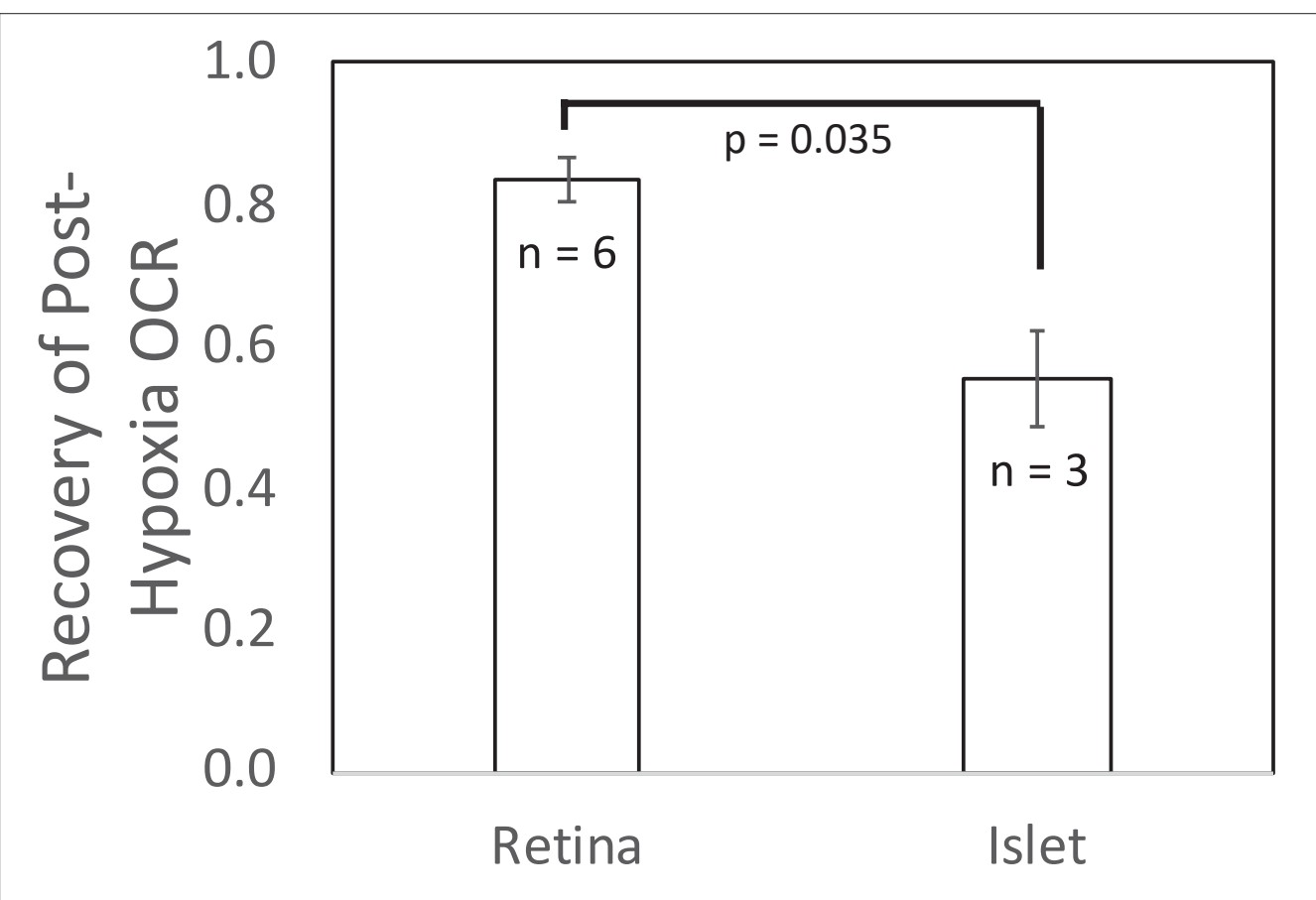

**Figure 5.** Recovery of $O_2$ consumption rate (OCR) after hypoxia relative to OCR prehypoxia values in retina and islets. Recovery data from experiments shown in ***Figures 2 and 4***. *t*-Test result: p = 0.035, for retina, *n* = 6, and for islets, *n* = 3. Raw data can be found in a Source Data file named '***Figure 5—source data 1***'.

The online version of this article includes the following source data for figure 5:

**Source data 1.** Recovery of oxygen consumption after hypoxia.

---

levels of $O_2$ that each tissue was exposed to, the supply gas tanks used for the hypoxia phase of the experiments were selected to generate similar outflow $O_2$ levels for the retina (1% yielded an outflow of 6.5 mm Hg) vs. islets (3% yielded 5.5 mm Hg).

In response to hypoxia, lactate and pyruvate production rates by retina increased, consistent with operation of the Pasteur effect. The ratio of lactate/pyruvate also increased during low $O_2$ conditions (*Figure 4B*), reflecting the decreased uptake of pyruvate into the mitochondria, and the increase in the cytosolic redox state (NADH/NAD) that occurs during low $O_2$ (*Lai and Miller, 1973*). These results provide support for the utility of measuring extracellular lactate and pyruvate for real-time responses to events affecting metabolism.

## Complex time- and concentration-dependent effects of H₂S on ISR by islets resolved by fluidics analysis

Past studies on the effect of $H_2S$ on ISR were consistent in their findings that $H_2S$ was inhibitory (*Niki and Kaneko, 2006*; *Wu et al., 2009*; *Yang et al., 2005*; *Bełtowski et al., 2018*; *Tang et al., 2013*; *Lu et al., 2019*; *Patel and Shah, 2010*; *Yang et al., 2007*). However, as $H_2S$ has both stimulatory and inhibitory effects on the ETC (*Khan et al., 1990*; *Vitvitsky et al., 2018*), we predicted that precise titration of the exposure of islets to $H_2S$ would reveal stimulatory effects of $H_2S$ on ISR. To simplify the analysis and interpretation, we ramped up the $H_2S$ concentration in the gas equilibration system (while the inflow and outflow gas ports were clamped) to accumulate $H_2S$ until the desired concentration was reached, and then clamped the permeation tube inlet port to maintain that $H_2S$ concentration for the indicated times. When $H_2S$ was increased until a steady state of 240 µM was reached, ISR from pancreatic islets increased by 35% relative to ISR at 20 mM glucose (*Figure 6A*). The increased ISR was sustained for 3 hr. The effect of $H_2S$ was reversible. After purging it from the system, ISR rapidly returned to levels that occurred prior to $H_2S$ exposure. In the presence of 3 mM glucose, $H_2S$ had no effect on ISR (data not shown), supporting the idea that this reflects a physiologic response of ISR to $H_2S$. To demonstrate the ability of the flow system to more fully characterize the time- and concentration dependency of ISR on $H_2S$, we measured ISR at steady-state concentrations of $H_2S$ from 80 to 780 µM. Notably, between 180 and 780 µM $H_2S$ (*Figure 6B*), the initial period of stimulation of ISR (peaking between 1 and 1.5 hr after the start of the ramp of $H_2S$) was insensitive to the concentration of $H_2S$. In contrast, the effect of higher levels of $H_2S$ inhibited the ISR rate only between 1.5 and 4 hr following the start of the $H_2S$ exposure. The initial upslope of ISR occurred with a delay of about 30 min, and both the stimulation and inhibition of ISR were rapidly reversible following the washout of $H_2S$ (*Figure 6B*). The ability of the system to resolve the time lag for $H_2S$ to activate ISR is limited by the rate of the increase of $H_2S$ in the gas equilibration system accomplished by the permeation tube. In order to increase the temporal resolution, the permeation tube leak rate can be increased, or the volume of the gas equilibration system must be decreased. Additional concentrations of $H_2S$ were tested and the observed peak of ISR, and the steady-state level between 3 and 4 hr were plotted as a function of the concentration of $H_2S$ (*Figure 6C*), clearly showing the ability of the system to resolve both the time courses and concentration dependency of the effects of a relatively small range of [$H_2S$]. In order to demonstrate the ability of the system to investigate mechanisms mediating the effects of $H_2S$ on ISR, we subsequently measured intracellular $Ca^{2+}$ in response to $H_2S$ at a concentration that increased ISR (*Figure 6D*). Consistent with a stimulatory effect of $H_2S$ on ISR, intracellular $Ca^{2+}$ increased dramatically upon exposure to $H_2S$, and returned to near pre-$H_2S$ levels within a few minutes of washing out the $H_2S$.

To test the assumption made in many studies that due to rapid equilibrium between $HS^-$ and dissolved $H_2S$, NaHS is able to emulate direct exposure to $H_2S$ (*Li and Lancaster, 2013*), we also analyzed the effect of NaHS on ISR and $Ca^{2+}$ by perifused islets. In contrast to dissolved gaseous $H_2S$, low levels of NaHS had no effect, and higher concentrations (>1 µM) inhibited ISR (*Appendix 1— figure 1A, B*) – consistent with findings of all previous studies that used NaHS as an $H_2S$ surrogate (*Niki and Kaneko, 2006*; *Wu et al., 2009*; *Yang et al., 2005*; *Bełtowski et al., 2018*; *Tang et al., 2013*; *Lu et al., 2019*; *Patel and Shah, 2010*; *Yang et al., 2007*). Similarly, NaHS did not increase $Ca^{2+}$, and slightly decreased it by amounts that were relatively insensitive to the concentration of NaHS (*Appendix 1—figure 1C*). The inhibitory effects of $HS^-$, which is present at about double the concentration of dissolved $H_2S$, may in part account for the complex concentration dependency seen in *Figure 6B, C*. The likely explanation of the difference between effects of $H_2S$ and NaHS is that the

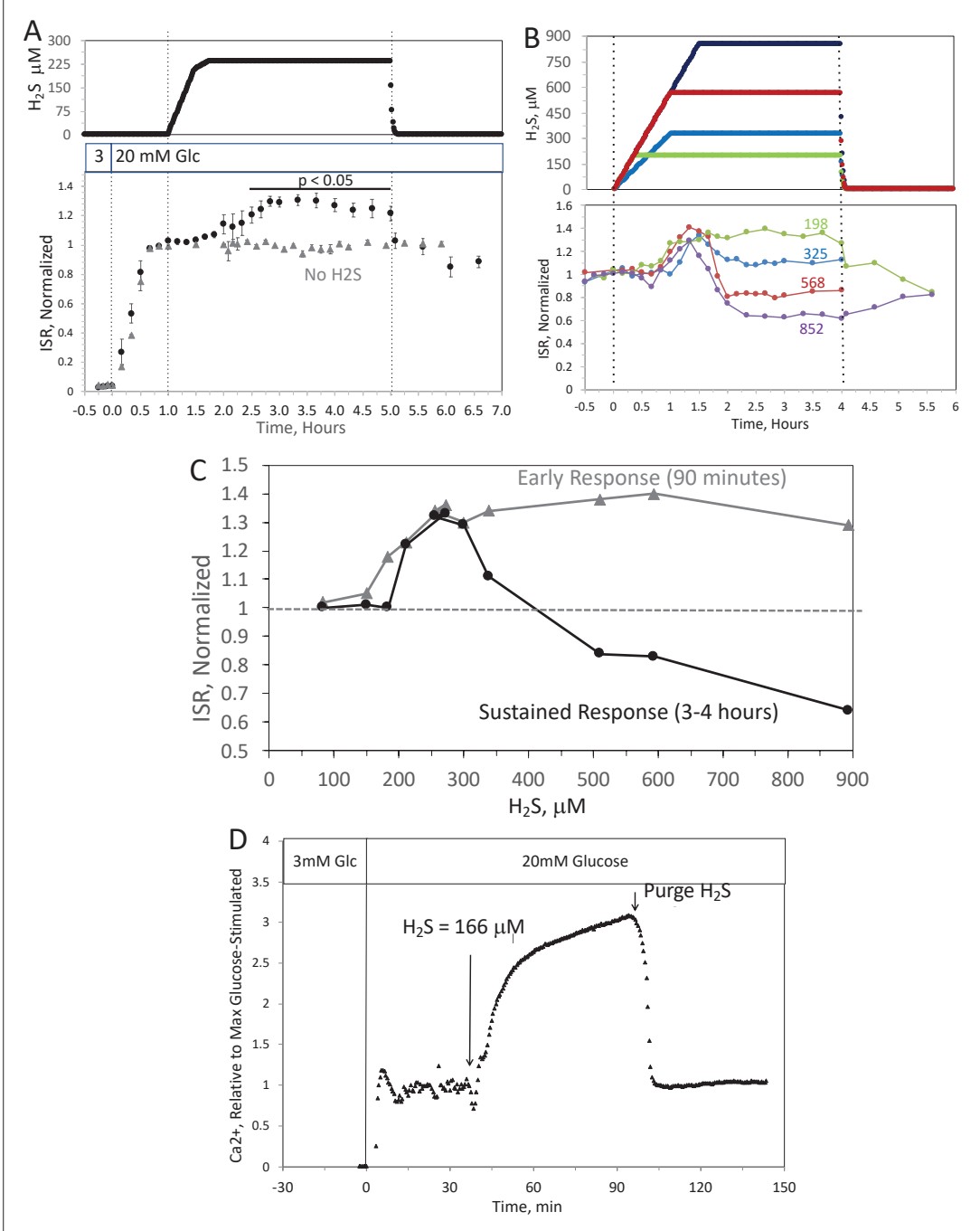

**Figure 6.** Effect of $H_2S$ on insulin secretion rate (ISR) by islets. (**A**) Rat islets (50/channel) were perifused (flow rate = 200 µl/min), and ISR was measured in response to glucose and exposure to dissolved $H_2S$ in the concentrations shown (data are average ± standard error [SE], $n = 3$ [$H_2S$], $n = 2$ [no $H_2S$], $p < 0.05$ as indicated). (**B**) ISR was measured at the indicated concentrations of dissolved $H_2S$. Each curve is a single experiment. (**C**) Data from perifusions as shown in B were plotted as a function of the ISR at the peak between 1 and 1.5 hr, and the average ISR between 3 and 4 hr. (**D**) Response of cytosolic $Ca^{2+}$ to changes in glucose concentration and exposure to 166 µM $H_2S$ and its washout. Raw data can be found in a Source Data file named '*Figure 6—source data 1*'.

The online version of this article includes the following source data for figure 6:

**Source data 1.** Effect of hydrogen sulfide on insulin secretion rate.

$H_2S$ generated by NaHS diffuses out of solution and into the gas phase when the media is in contact with a headspace that does not contain gaseous $H_2S$. To test this, 10 ml of KRB were placed in a sealed 125 ml bottle, and measured $H_2S$ in the KRB and the headspace as a function of time after either injecting NaHS into the solution or permeating gaseous $H_2S$ into the headspace (*Appendix 1—figure 2A, B*). The $H_2S$ in solution emanating from the headspace was higher than that achieved from NaHS, but notably it declined when the headspace was purged. Although the amount in the headspace was close to the detection limit of the $H_2S$ measurement method, it was clear from the sharp decline in $H_2S$ after unsealing the bottle, that $H_2S$ from the NaHS had indeed transferred into the headspace. Thus, these data support a scenario where differences in effects of $H_2S$ and NaHS occur due to the absence of $H_2S$ in solution containing NaHS and $HS^-$ is inhibitory for both ISR and $Ca^{2+}$.

## Measurement of OCR, cytochrome c, lactate, and pyruvate by perifused liver slices

We also explored the ability of our flow system to measure effects of $H_2S$ on liver by measuring OCR, cytochromes, and lactate/pyruvate release by liver slices in the absence and presence of a mitochondrial fuel (succinate). The responses were complex, changed directions in time- and concentration-dependent fashion, and will ultimately require more experiments to interpret the data mechanistically. Therefore, these data were placed in the appendix (*Appendix 1—figures 3 and 4*). Nonetheless the waveforms of the responses were clearly resolved, confirmed the capabilities of the multiparametric detection system, and so were included in this report. The salient features of the data can be summarized by: (1) In the absence of the mitochondrial fuel succinate, $H_2S$ changed OCR and reduced cytochromes in proportion to each other, consistent with donation of electrons from $H_2S$ to cytochrome c (*Vitvitsky et al., 2018*; *Appendix 1—figure 3*); (2) In the presence of a TCA cycle intermediate (succinate), $H_2S$ (200–300 µM) increased the reductive state of cytochrome c oxidase while decreasing OCR. This is consistent with inhibition of cytochrome c oxidase (*Khan et al., 1990*; *Appendix 1—figure 4*), which occurred at concentration similar to typical estimates of plasma concentration which range from 30 to 300 µM (*Olson, 2009*). At low levels, $H_2S$ caused irreversible inhibition of ETC activity upstream of cytochrome c but did not inhibit flow of electrons from succinate. Thus, both reported mechanisms of action of $H_2S$ on the ETC were resolved by the system, as well as uncovering additional effects of $H_2S$ that had not previously been reported.

## Discussion

### General features of the flow system

Flow systems have important advantages over static systems for assessment of cell function. Viability and functions of cells and tissues are better and closer to physiological, culture media composition can be changed, and real-time production or uptake rates can be quantified from differences between inflow and outflow. Microfluidics devices can maintain tissues in ways that preserve their three-dimensional structure and preserve native cell-to-cell interaction (*Nieskens and Wilmer, 2016*; *Rothbauer et al., 2018*). However, these devices will have maximal impact when combined with real-time assessment of the tissue as well as the ability to control aqueous and gaseous composition of the media bathing the tissue models. This report focuses on technical modifications to a previously developed flow culture/assessment system (*Sweet et al., 2004*) that enables real-time measurements of responses of tissues to physiologically important dissolved gases. We achieved this by incorporating a unique gas equilibration system that controls abundant (blood) gases including $O_2$, $CO_2$, and $N_2$, and by using permeation tubes to introduce and control endogenously-produced gases such as $H_2S$, NO, and CO. In this report, we demonstrated the utility of this system using both a blood ($O_2$) and a signaling gas ($H_2S$).

### Control and effects of dissolved $O_2$: Recovery of metabolic state following hypoxia and transient response in OCR following reoxygenation

The ability to control dissolved $O_2$ makes our system highly suitable for investigating ischemia–reperfusion injury, generating two informative endpoints from a protocol that measures effects of a short period of low $O_2$ availability followed by return to normal levels. We used the recovery of OCR and

cytochrome c reduction to report tissue sensitivity to hypoxia; and we identified a transient spike in OCR that occurs upon reintroduction of normal $O_2$ levels.

The first endpoint characterizes the capacity of a tissue to survive after exposure to selected time periods of low $O_2$. In the illustration carried out in this study, retina recovered to 83% after hypoxia, whereas islets recovered to only 55% of prehypoxic levels of OCR, corresponding to the two tissue's known sensitivity to oxidative stress. The second endpoint, the burst of OCR occurring when $O_2$ floods back into the cell, is one that has been hypothesized, but has not previously been measured due to the difficulty of measuring OCR in the face of changing $O_2$ levels. Our method has enabled the measurement of transient responses to reoxygenation and revealed a two-phase waveform in OCR in both islets and retina. The ability to resolve this waveform was dependent on rigorous convolution analysis to remove the delay and dispersion of the $O_2$ signal due to the flow system, combined with ultra-stable and ultra-sensitive $O_2$ sensors. It has been long recognized that ROS is generated rapidly by cells when $O_2$ becomes plentiful after undergoing hypoxic conditions (*Chouchani et al., 2016*; *Zweier et al., 1987*). The transient spike of OCR is consistent with a scenario where $O_2$ is the source of oxygen atoms for the ROS. This is also consistent with the buildup of metabolites such as succinate during ischemia (*Beach et al., 2020*), which could fuel increased oxidation rate. The detailed relationship between OCR and generation of ROS will be the topic of future applications with this system. For instance, the method can be used to test whether slowed reintroduction of $O_2$ or candidate therapeutics including $H_2S$, prevent or reduce the transient spike in OCR while at the same time preventing the decreased recovery following the hypoxic period (as has been hypothesized; *Clark and Gewertz, 1992*; *Zhang, 2020*; *Citi et al., 2018*; *Du et al., 2017*). Other applications could include: testing whether there are differences in transient OCR response for different tissues or metabolic states; determining the relation of the spike to the response of ROS and recovery/survival of tissue; and testing drugs designed to prevent both the transient spike and/or the decrease in posthypoxic OCR. The ability to objectively quantify recovery of OCR positions the system to be used to test tissue sensitivity to a wide range of stresses (including ER and oxidative stress, immunological stress [exposure to cytokines], lipotoxicity as well as hypoxia), and to test drugs and treatments designed to increase or decrease recovery from experimentally induced stresses.

## Control and effects of a trace gas: stimulation of ISR and Ca$^{2+}$ by H$_2$S in islets

Gas signaling molecules (CO, NO, and $H_2S$) are generated in most tissues and have wide-ranging effects on function, metabolism, and protection from hypoxia (for reviews *Ahmad et al., 2018*; *Beltowski, 2007*; *Cheng and Rong, 2017*; *Krylatov et al., 2021*). However, due to the difficulty in quantitatively and reproducibly introducing dissolved trace gases into culture media, the majority of studies on these gases have utilized aqueous chemical donors such as NaHS instead of $H_2S$. These surrogates provide only imprecise and uncertain concentrations and timing of tissue exposure to dissolved $H_2S$. Due to the extremely accurate calibration of the rate of release of gases, permeation tubes can introduce trace gases into the carrier gas (the mixture of $CO_2$–$O_2$–$N_2$) present in a gas equilibration system of our perifusion apparatus at exact times and concentrations. Producers of permeation tubes can provide them with a selection of over 500 gases, including $H_2S$, NO, CO, and ammonia. Thus, the method of incorporating permeation tubes into the gas equilibration system is particularly versatile and can be used for a wide range of applications.

To illustrate the unique advantages of being able to expose tissue to precise levels of dissolved $H_2S$, we selected islets as a test tissue since the inhibitory effects of $H_2S$ on glucose-stimulated ISR and cytosolic Ca$^{2+}$ by islets have been described. However, those reports were based on use of the $H_2S$ donors NaHS and Na$_2$S (*Niki and Kaneko, 2006*; *Wu et al., 2009*; *Yang et al., 2005*; *Beltowski et al., 2018*; *Tang et al., 2013*; *Lu et al., 2019*; *Patel and Shah, 2010*; *Yang et al., 2007*) and the interpretation that $H_2S$ gas is actually delivered to the tissue and at levels low enough to avoid its toxic effects. Moreover, we took note of other studies that suggested that $H_2S$ can donate electrons directly to cytochrome c (*Vitvitsky et al., 2018*) and there is evidence that the reduction of cytochrome c may be a key regulatory step in activating ISR (*Rountree et al., 2014*; *Jung et al., 2011*). Our system bore out this prediction revealing stimulatory effects of $H_2S$ on ISR that had not been apparent when exposing islets to a donor of $H_2S$ (NaHS). $H_2S$ at concentrations (between 140 and 280 µM) enhanced glucose-stimulated ISR, which remained elevated for at least 4 hr. The

concentration- and time dependency were complex however. At higher concentrations of $H_2S$ the stimulation of ISR for 90 min still occurs, but at later times ISR was inversely proportional to the $H_2S$ ranging from a 35% increase to a 40% decrease. The U-shaped concentration dependency may also reflect the effects of both $H_2S$ and inhibitory effects of $HS^-$. At low concentrations NaHS had little effect on ISR or cytosolic $Ca^{2+}$, however as its concentration approached 1 µM and above, both endpoints were inhibited. The range of effects occurred over a relatively small range of $H_2S$ concentrations, highlighting the need for the very precise control of dissolved $H_2S$ afforded by the use of permeation tubes to investigate this phenomenon. $H_2S$ had no effect on ISR at 3 mM glucose, supporting a physiologic mechanism mediating $H_2S$'s effect that may be integrated with glucose sensing and secretory response to fuels by the islet (*Prentki et al., 2013*; *Campbell and Newgard, 2021*).

The stimulatory effects of $H_2S$ on ISR and $Ca^{2+}$ directly refute the well accepted conclusions that $H_2S$ decreases ISR and $Ca^{2+}$ by opening $K_{ATP}$ channels as has been widely reported (*Ali et al., 2007*; *Yang et al., 2005*; *Lu et al., 2019*; *Shoji et al., 2019*). Although it is typically assumed when using NaHS as a source of $H_2S$ that NaHS and $H_2S$ equilibrate in solution (*Olson, 2009*), our measurements show that in the absence of $H_2S$ in the headspace above the media, dissolved $H_2S$ quickly disappears into the headspace, a phenomenon analogous to the behavior of $CO_2$-based buffers. Thus, measurements of effects of NaHS on tissue in open systems such as typical static and perifusion methods contain little $H_2S$. Thus, our method offers a novel and uniquely capable approach for investigating the direct effects of the protonated form of $H_2S$ in equilibrium with $HS^-$.

$H_2S$ is generated in islets by the action of three intracellular enzymes (*Kimura, 2010*) but is also a component of blood albeit at levels that are not well established (*Whiteman et al., 2010*; *Karunya et al., 2019*; *Whitfield et al., 2008*). It is notable that the range of concentrations that induced changes in ISR, and above which caused inhibition of OCR in liver are in the range of typical estimates of plasma concentration which range from 30 to 300 µM (*Olson, 2009*). Therefore, it suggests that intracellular effects of $H_2S$ could be mediated by $H_2S$ derived from the blood as well as $H_2S$ endogenously produced by cells. The ability to detect differences between responses to $H_2S$ and donor molecules, will be useful to the increasing numbers of investigators developing $H_2S$ donor molecules as pharmaceutics (*Citi et al., 2020*; *Testai et al., 2020*). The increase in ISR in response to $H_2S$ has physiological, methodological, and clinical implications and the lack of similar stimulatory effects by a donor molecule has broad implications in a field where studies of NO, $H_2S$, and CO are mostly based on the use of donor molecules.

## Lactate and pyruvate: relation to cytosolic events

The assayed values of lactate and pyruvate reflect a number of important specific and global parameters. The rate of release of lactate and pyruvate is an integration of the rate of glycolysis less the amount of pyruvate flux into the mitochondria and traversing gluconeogenesis. Thus, both compounds generally increase in response to glycolytic fuels. Importantly for the study of hypoxia, both metabolites rise in cells when $O_2$ is decreased (the Pasteur effect). In addition, the ratio of cytosolic lactate/pyruvate mirrors the cytosolic NADH/NAD ratio due to the equilibrium status of the LDH reaction (*Newsholme and Start, 1973*). One could envision that freeze clamping cells and measuring intracellular lactate and pyruvate to directly compare them to the extracellular values could be a way to validate the use of extracellular data. However, in practice, the measurement of intracellular compounds is difficult and also limited by the kinetic resolution of freeze clamping. Instead, we evaluated the responses of the extracellular levels of lactate to a blocker of LDH (oxamate), of pyruvate to a blocker of mitochondrial transport (zaprinast), and both compounds in response to hypoxia. The rapid changes in extracellular lactate and pyruvate support the rapid redistribution between intra- and extracellular compartments, and that real-time measurement of extracellular lactate and pyruvate reflect intracellular events governing intracellular lactate and pyruvate. Retina responded to hypoxia with a classical Pasteur effect: low $O_2$ increased lactate, pyruvate, and lactate/pyruvate ratio. We envision that the measurement would be especially informative when examining the shift from oxidative to glycolytic metabolism such as seen in tumorigenesis (*de Groof et al., 2009*) or stem cell differentiation (*Zhou et al., 2012*).

### Incorporation of $O_2$ control into a real-time fluorescent imaging system

Real-time fluorescent imaging is a powerful modality that is commonly used to quantify a wide variety of intracellular compounds and factors while perfusing the optical chamber housing the cells or tissue. Molecular probes provide intracellular dyes for over 100 separate compounds, so this method is versatile and wide ranging. Thus, incorporating the gas equilibration system to a flow system providing buffer to a chamber that images of single islets. We observed and quantified clear increases in intracellular $Ca^{2+}$ in response to hypoxia as metabolic rate decreased. Unexpectedly, the loss of energy and ISR following islet exposure to hypoxia were not accompanied by a loss of glucose-stimulated $Ca^{2+}$, suggesting the mechanism mediating loss of ISR is independent of $Ca^{2+}$ signaling. These data are consistent with previous findings that loss of secretory function is more closely associated with bioenergetics than $Ca^{2+}$ (*Rountree et al., 2014*; *Rountree et al., 2013*). When comparing results of various assays and modes of analysis, the ability to measure multiple endpoints under matched conditions and the same flow system is optimal for systematic study of tissue function.

### Summary of uses for the flow culture system

The ability to precisely control the levels and timing of exposure to both abundant and trace gases while measuring multiple parameters in real time on a wide range of tissue and cell models make this system uniquely powerful. Although we illustrated the method using rodent tissue, it will have particular utility when maintaining and assessing human tissue. The novel resolution of OCR transients attests to the high kinetic resolution of the system. Moreover, stimulatory effects of $H_2S$ on ISR and $Ca^{2+}$ not seen in response to donor molecules, attests to the ability of the technology to reveal behavior that provides new insight. Given the wide use of donor molecules in studying gasotransmitters this has broad and significant implications. In addition to enabling users to evaluate direct effects of gases, the method is also suitable for testing conditions or agents that diminish loss of OCR in response to hypoxia- or other stress-induced effects. The use of methods to study the effects of dissolved gas on tissue will impact many areas of fundamental research as well as research of diseases including but not limited to diabetic wound healing, stroke (ischemia/reperfusion injury), and cancer.

## Materials and methods
### Chemicals

Krebs–Ringer bicarbonate buffer was used for all perifusions prepared as described previously (*Neal et al., 2015*). Antimycin A, glucose, oxamate, KCN, and zaprinast were purchased from Sigma-Aldrich. Gases of varying $O_2$ levels/5% $CO_2$ and balance $N_2$ were purchased from Praxair Distribution Inc (Danbury, CT). Cytodex and Cytopore beads were purchased from GE healthcare and biosciences (cat no. 17-0448-01 and 17-0911-01, respectively).

### Culture of INS-1 832/13 cells

INS-1 832/13 cells were kindly provided by Dr. Christopher Newgard (Duke University) who initially generated the cell line (*Hohmeier et al., 2000*). Their identity was confirmed by a variety of functional tests that reveal characteristics unique to this beta cell line (increase in OCR, intracellular calcium, lactate production, and insulin secretion rate in response to physiological changes in glucose [5–10 mM glucose] that are intrinsic to beta cells). Tests for mycoplasma contamination in the cell suspension were negative. Cells were grown and cultured as previously described (*Hohmeier et al., 2000*). The day before experiments, cells were harvested, and cultured with Cytodex beads (2.5 mg/million cells) for 15 min in RPMI Media 1640 (Gibco, Grand Island, NY) supplemented with 10% heat-inactivated fetal bovine serum (Atlanta Biologicals, Lawrenceville, GA), 2 mM L-glutamine, 1 mM pyruvate, 50 µM beta-mercaptoethanol, 20 mM HEPES, and 1 % Pen/Strep. They were then washed and cultured overnight in a standard $CO_2$ incubator at 37°C.

### Tissue harvesting and processing

All procedures were approved by the University of Washington Institutional Animal Care and Use Committee.

## Rat islet isolation and culture

Islets were harvested from male Sprague-Dawley rats (approximately 250 g; Envigo/Harlan, Indianapolis, IN) anesthetized by intraperitoneal injection of sodium pentobarbital (150 mg/kg rat) and purified as described (**Sweet et al., 2004**; **Matsumoto et al., 1999**). Subsequently, islets were cultured for 18 hr in RPMI Media 1640 supplemented with 10% heat-inactivated fetal bovine serum (Invitrogen) at 37°C prior to the experiments.

## Retina isolation

Retinas were harvested from C57BL/6J mice (euthanized by cervical dislocation) 10 min prior to loading and were dissected into ¼ths using microscissors as previously described (**Bisbach et al., 2020**).

## Flow culture system to maintain tissue with precise control of dissolved gases

A flow culture system (**Neal et al., 2015**) was modified to continuously perifuse tissue with buffer equilibrated with the desired composition of dissolved gas using a gas equilibration system (**Figure 7A, B**). Multiple modes of assessment were integrated into the flow culture system and are described below including chemical sensors for $O_2$, spectroscopic analysis of the tissue for measurement of reduced cytochrome c and cytochrome c oxidase, and collection of outflow fractions for subsequent assay of lactate and pyruvate, or insulin. Model numbers and manufacturers are listed in the legend for **Figure 7**. Prior to entering the perifusion chamber, perifusate is pumped from the media reservoirs by an eight-channel peristaltic pump into the thin-walled silastic tubing of the gas equilibration system that facilitated equilibration between the buffer and gas in the glass housing. The use of the gas equilibration system avoids the issue of outgassing of dissolved gases (**DeLeon et al., 2012**) from the perifusate prior to flowing past the tissue and sensors. Selection of media reservoir with desired media composition determined by use of a six-port valve. To achieve the desired gas composition of $O_2$, $CO_2$, and $N_2$ within the gas equilibration system, tanks of premixed gases supplied gas to the inflow port, typically 5% $CO_2$, the desired percentage of $O_2$, and balance $N_2$.

To equilibrate the inflow with desired concentration of $H_2S$, we have used devices called permeation tubes (VICI Metronics, Poulsbo, WA), an industry standard that provides very precise rates of gas release, typically from 1 to 30 µg/min. The outlet of the permeation tube was connected to the inlet port of the chamber housing the gas equilibration system (**Figure 7C**), so that the headspace around the perifusate in the gas-permeable tubing accumulated the trace gas. The concentration in the chamber then increased as a ramp function, rising at a rate equal to the leak rate of the permeation tube x time divided by the volume of the housing. Henry's constant is defined as

$$Hc = 1000 \times [gas]_{aq}/[gas]_g$$

where $[gas]_{aq}$ is in µM, and $[gas]_g$ is in ng/ml.

At 37 degrees dissolved $O_2$ is 217 nmol/mL in KRB, which is in equilibrium with 0.3008 mg/mL of O2 in air (21%). Therefore,

$$Hc(O_2) = 1000 \times [O_2]_{aq}/[O_2]_g = 217\mu M/300,800ng/mL \times 1000 = 0.721$$

In order to estimate Henry's constants for $H_2S$ at 37 degrees in KRB, we used measurement of solubility (reported by National Institute of Standards and Technology) normalized relative to $O_2$ in the relationship

$$Hc(H_2S) = Solubility(H_2S)/Solubility(O_2) \times Hc(O_2) = 0.1/0.0013 \times 0.721 = 55.5$$

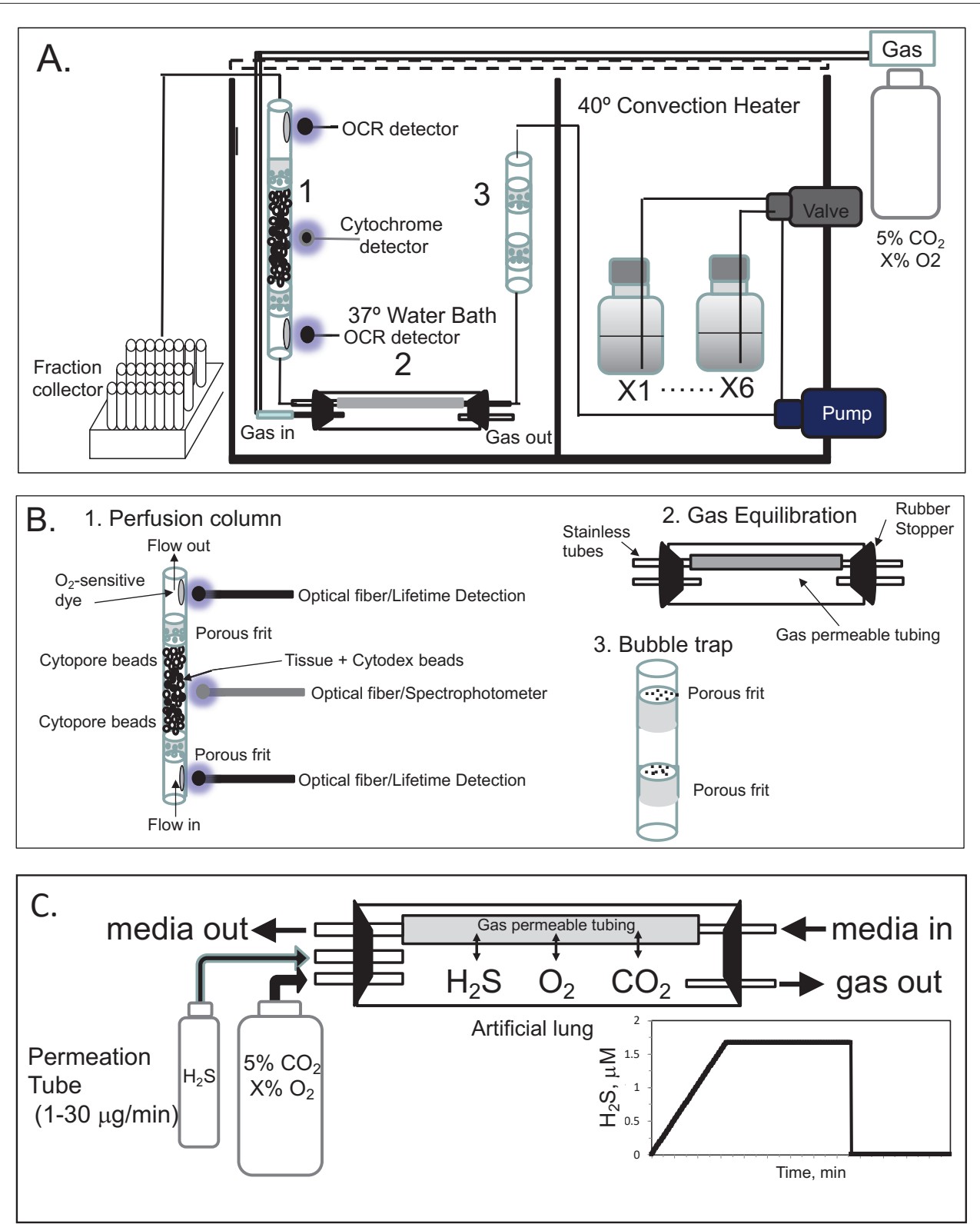

**Figure 7.** Flow culture system/assessment system for studying effects of dissolved gases on tissue or cells. One channel/perifusion chamber is shown, but the actual system can accommodate up to 8. (**A**) The perifusion system consisted of an eight-channel peristaltic pump (MiniPuls 2, Gilson, Middleton, WI) connected to a six-port valve (Part # V-451, IDEX Health and Science, Oak Harbor, WA) to produce up to six separate solutions; a media/dissolved gas equilibrium system in which media flowed through thin-walled silastic tubing (0.062 in ID × 0.095 in OD; Dow Corning Corp, Midland, MI)

*Figure 7 continued*

for a residence time of 5 min (typically 0.2–0.5 m depending on the flow rate) loosely coiled in a glass jar that contained various $O_2$, 5% $CO_2$/balance $N_2$; bubble trap comprised of a Simax Borosilicate glass tube (Mountain Glass, Asheville, NC, 2" long and 4.2 mm ID) filled with glass wool; a Simax Borosilicate glass perifusion chamber (3" long and 4.2 mm ID) immersed in a 37°C water bath; Lifetime detection spectrometers (Tau Theta, Boulder CO), a tungsten-halogen light source/USB2000 spectrophotometer (Ocean Optics OH); and a Foxy 200 fraction collector (Isco, Inc, Lincoln, NE). (**B**) A blow up of individual parts shown in A. (1) The glass perifusion chamber containing culture beads, porous frits (Interstate Specialty Products, Suton, MA, Cat no. POR 4894, cut to 4.2 mm diameter and 0.25 in long) to support the tissue and disperse the flow, and coated with $O_2$-sensitive dye on the interior above and below where the tissue resides; (2) gas equilibration chamber, where media flows through gas-permeable silastic tubing and equilibrates with the gases filling the headspace; (3) bubble traps. (**C**) Incorporation of a permeation tube (VICI Metronics, Poulsbo, WA) that releases $H_2S$ at specified rates into the media/dissolved gas equilibration system during which time the ports for the inflow and outflow of carrier gas ($O_2$, $CO_2$, and $N_2$) are closed. The resulting accumulation of $H_2S$ in the artificial lung yields linearly increasing concentrations of dissolved $H_2S$ in the form of a ramp function as shown.

Thus, the equation relating the dissolved $H_2S$ concentrations to head space concentration is:

$$[H_2S_{aq}] = [H_2S_g] \times \ Hc/1000 \tag{1}$$

where $[H_2S_{aq}]$ is in µM, and $[H_2S_g]$ is in ng/ml.

## Lifetime detection of dissolved $O_2$

$O_2$ tension in the inflow and outflow buffer was measured by detecting the phosphorescence lifetime of an $O_2$-sensitive dye painted on the inside of the perifusion chamber using a MFPF-100 multifrequency phase fluorometer lifetime measurement system (TauTheta Instruments, Boulder, CO) as previously described (*Sweet et al., 2002*). Using tanks of gas containing varying amounts of $O_2$ (21%, 15%, 10%, 5%, 3%, 1%, or 0%), data were generated that showed the dependency of the lifetime signal as a function of $O_2$ and the rapidity of changes in $O_2$ after each change in gas tank. Within 5 min, $O_2$ achieves 95 % of steady-state levels (*Figure 8A*), where the delay is primarily due to time needed for the gas in the gas equilibration system to turnover as the actual sensor responds in microseconds. The $O_2$ dependency of the dye signal conformed to the Stern–Volmer equation.

$$Lifetime = 1/(k_1 + k_q * [O_2]^{1/2}) \tag{2}$$

where lifetime is in µs. *Equation 2* was used to as a calibration curve to convert the optical signals to $O_2$ content (*Figure 8B*). The use of lifetime detection produces very stable and sensitive data at both normal (*Figure 8C*) and low (*Figure 8D*) $O_2$ levels producing S/N over 20 even when measuring a change of only 1.9 mm Hg.

## Continuous measurement of OCR

### Measuring the difference between inflow and outflow during invariant inflow $O_2$

When inflow $O_2$ tension is constant, OCR by the tissue equals the difference between the content of $O_2$ flowing into the perifusion chamber minus that flowing out time the flow rate as follows.

$$OCR = ([O_2]in - [O_2]out) \times flowrate \tag{3}$$

where flow rate is in µl/min and $[O_2]$ is in nmol/ml. Inflow and outflow $O_2$ sensors were positioned on the inside of the perifusion chamber 2 cm upstream and 2 cm downstream from the tissue, respectively. Perifusate flow rates were set to result in a difference between inflow and outflow $O_2$ of between 5% and 25% of the inflow $O_2$ signal, so it was large enough to be accurately measured, but small enough to avoid exposure to unintended hypoxic conditions.

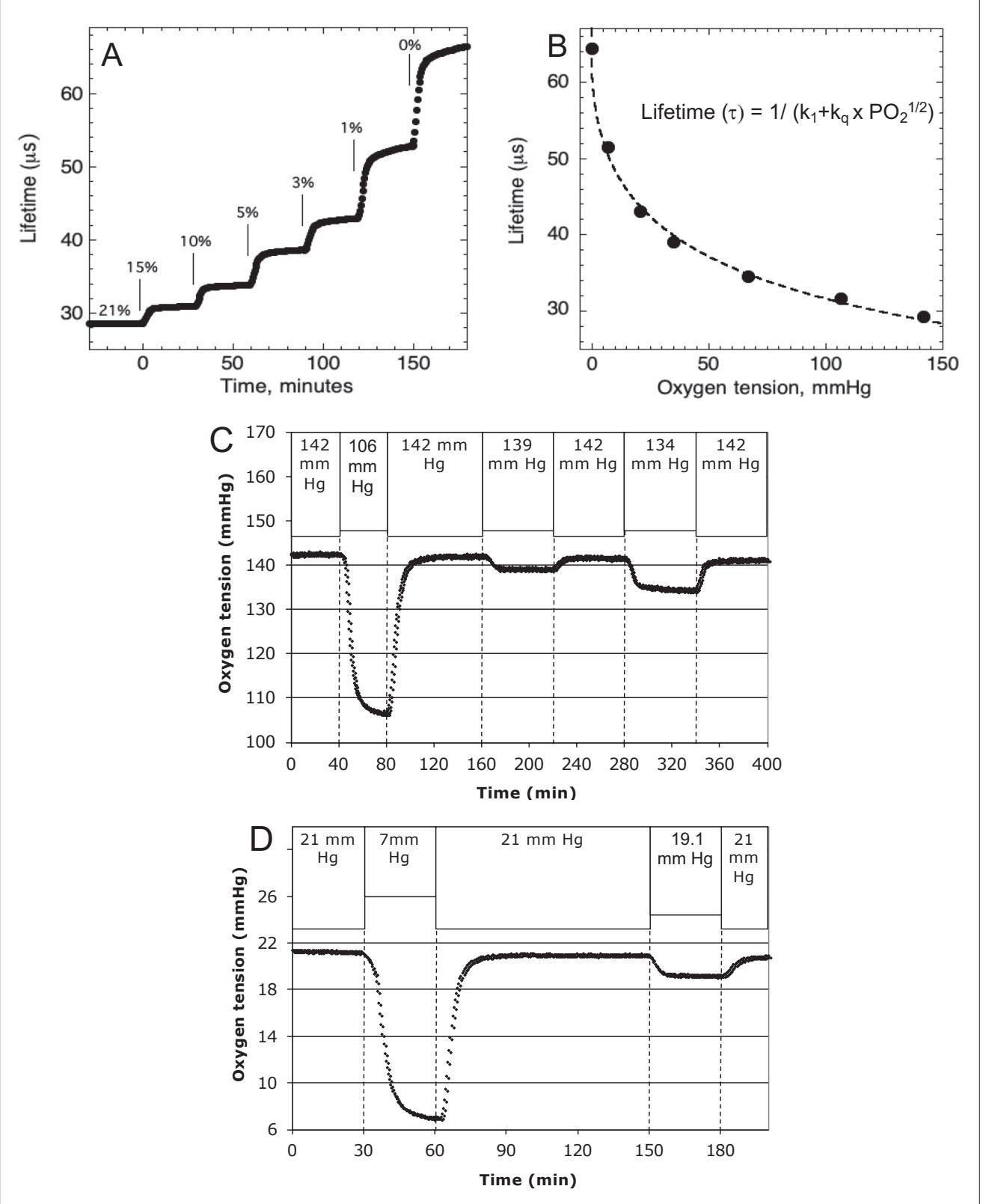

**Figure 8.** Control and measurement of dissolved $O_2$. (**A**) At 20- to 30-min intervals, the lung was sequentially filled with 21%, 15%, 10%, 5%, 3%, 1%, and 0 % $O_2$. (**B**) The steady-state lifetime measurements were nonlinear and conformed to a Stern–Volmer equation. Where the $pO_2$ is partial pressure of oxygen molecules (oxygen tension), $\tau_0$ is lifetime of the dye without quencher such as oxygen molecule, $k_1$ is $1/\tau_0$ and $k_q$ is a bimolecular quenching constant of the dye by oxygen molecules. The $k_1$ and $k_q$ were 15,519 s$^{-1}$ and 1612.2 mm Hg$^{-1}$ s$^{-1}$, respectively. (**C**) Test of precision at normal $O_2$. $O_2$

*Figure 8 continued on next page*

*Figure 8 continued*

(142 mm Hg) in the lung was changed first to 106.5 mm Hg and back to 142 (levels of change that were typical what was observed in our studies). By mixing 21% and 15% tanks at known flow rates, $O_2$ was then decreased by 3 mm Hg and then subsequently by 8 mm Hg. (**D**) Test of precision and S/N of low $O_2$. $O_2$ (21 mm Hg) in the lung was changed first to 7 mm Hg (1% $O_2$) and back to 21. By mixing 3% and 1% tanks at known flow rates, $O_2$ was decreased by 1.9 mm Hg. S/N was for the 14 and 1.9 mm Hg changes was >80 and >10, respectively. Raw data can be found in a Source Data file named '***Figure 8—source data 1***'.

The online version of this article includes the following source data for figure 8:

**Source data 1.** Control and measurement of dissolved $O_2$.

## Measuring OCR during changes in inflow $O_2$: convolution analysis to remove system effects

Measuring the temporal changes in OCR by tissue in the face of changing inflow concentrations of $O_2$ requires a correction for the difference in inflow and outflow $O_2$ levels due to the delay and dispersion generated between the inflow and outflow sensors. To calculate OCR from *Equation 3* in the face of changing inflow $O_2$, the inflow $O_2$ content must be converted to what it would be if the sensor was located at the outflow sensor location. This was done with classical convolution methods (*Weigle et al., 1987*) with mild regularization (*Bube and Langan, 2008*) to create a mathematical function representing the delay and dispersion of the inflow signal by the flow through the perifusion chamber from the inflow to the outflow sensor described numerically by *Equation 4*.

$$[O_2]_{in:transformed} = [O_2]_{in} * h(t) \tag{4}$$

where $[O_2]_{in:transformed}$ is the inflow concentration at the outflow sensor, and $h(t)$ is the system transfer function. In the absence of tissue in the flow system, $O_2$ was decreased to hypoxic levels in the same protocol as was done in the presence of tissue, while measuring $[O_2]$ in the inflow and outflow (*Figure 9*). The transfer function $h(t)$ was then generated for each experimental condition by solving *Equation 4* by deconvolution using MatLab. For each perifusion analysis, the measured $[O_2]_{in}$ was converted to $[O_2]_{in:transformed}$ by convolution with the transfer function (also using MatLab) and OCR was calculated from.

$$OCR = ([O_2]_{in:transformed} - [O_2]_{out}) \times FR \tag{5}$$

For the protocols in this experimental setup, only 35 min of the transfer function was needed to accurately transform the inflow $[O_2]$.

## Measurement of cytochrome c and cytochrome c oxidase reduction

The reductive states of cytochrome c and cytochrome c oxidase were quantified by measuring spectra of light transmission from 400 to 650 nm through the column of islets or tissue as previously described (*Chance and Williams, 1956a*; *Jung et al., 2011*). From these spectral data, absorption, first and second derivatives were calculated as described below. In contrast to other methods used to measure changes in heme redox state in spinner systems (*Kim et al., 2012*; *Kashiwagura et al., 1984*), our use of a flow system allows continuous measurement of cytochromes in tissue for extended periods of time where the tissue is exposed to controlled media composition and no mechanical damage is inflicted on tissue by the spinner. Due to the low signal to noise and baseline shift during the experiments, direct measurement of absorption was not stable. To better resolve changes in the reduced state of cytochromes the second derivative of the absorbance spectra with respect to wavelength was calculated (*Cavinato et al., 1990*). Like absorption, this parameter reflects the number of electrons bound to the cytochrome as well as the amount of protein. However, the second derivative is unaffected by shifts in baseline allowing resolution of real-time changes in absorbance. At the conclusion of each experiment, calibration spectra for fully oxidized and reduced cytochromes were acquired in the presence of blockers of the ETC – namely 12 µg/ml antimycin to stop the flow of electrons to cytochromes, followed by 3 mM KCN to facilitate the maximal accumulation of electrons bound to cytochromes.

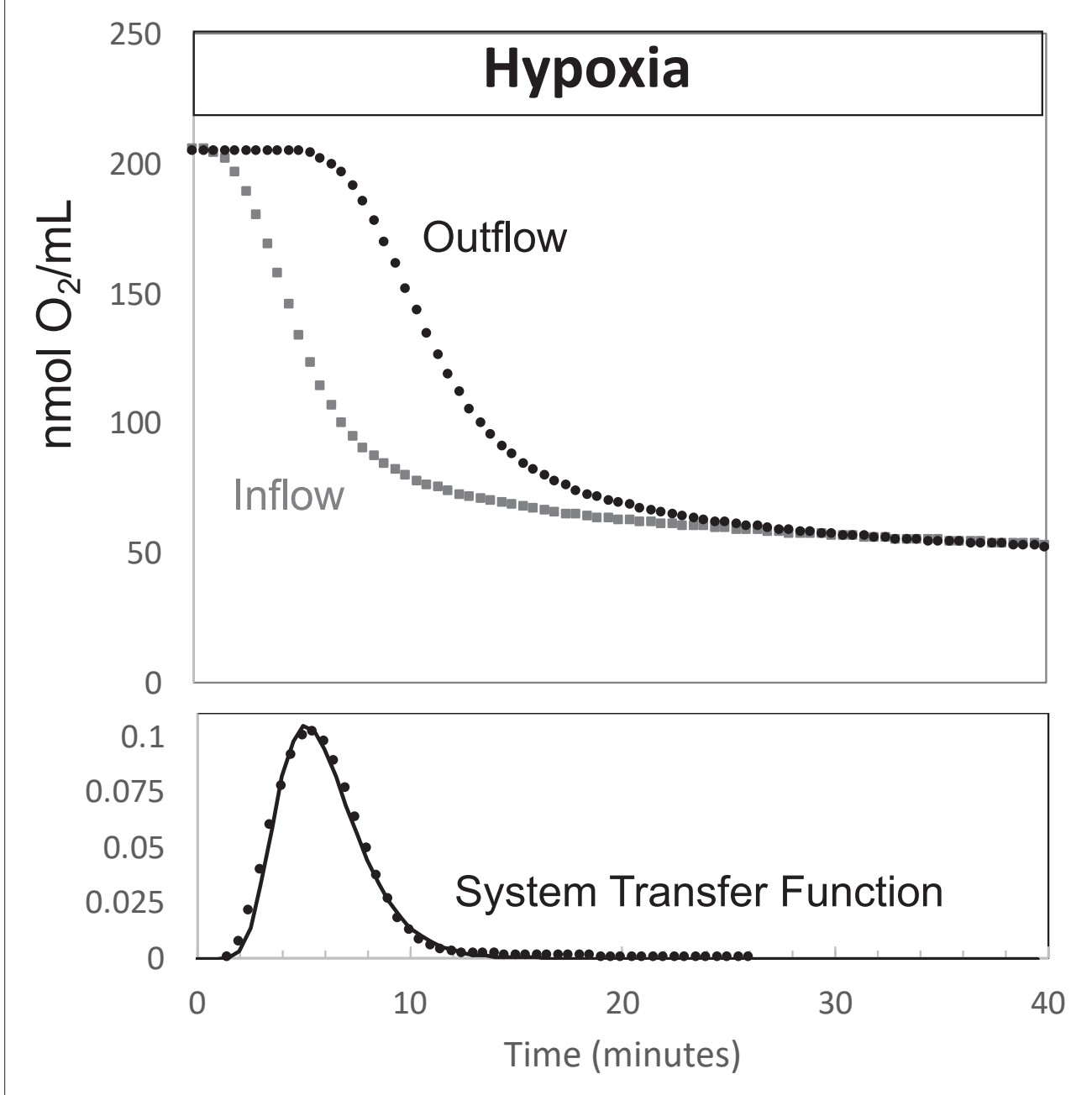

**Figure 9.** Determination of transfer function from inflow and outflow for convolution analysis. Measurement of inflow and outflow $O_2$ tensions in response to a change from 21% to 3% with no live tissue in the system. Deconvolution was carried out to generate the transfer function of the system shown on the bottom graph. Raw data can be found in a Source Data file named '*Figure 9—source data 1*'.

The online version of this article includes the following source data for figure 9:

**Source data 1.** Determination of transfer function from inflow and outflow data.

### Spectral data processing

The second derivative of absorbance with respect to wavelength (*Cavinato et al., 1990*) at 550 and 605 nm was calculated (corresponding to cytochrome c and cytochrome c oxidase, respectively) as,

$$Abs'' = \frac{\Delta\left(\frac{\Delta ABS}{\Delta V}\right)}{\Delta V} \tag{6}$$

where Abs = log (intensity − intensity$_{bkg}$)/(intensity$_{ref}$ − intensity$_{bkg}$), $v$ = wavelength in nanometers, and Δ = change in the variable over the integration interval. Background intensity (intensity$_{bkg}$) was determined with the light source off, and the reference intensity (intensity$_{ref}$) was that obtained when cytochromes were fully oxidized by antimycin A. Percent reduction of cytochromes were calculated following Kashiwagura et al. (*Kashiwagura et al., 1984*) as,

$$Cytochrome_{red} = 100 \times \frac{Abs'' - Abs''_{aA}}{Abs''_{KCN} - Abs''_{aA}} \tag{7}$$

where Abs″ and Abs″$_{KCN}$ are values at 550 or 605 nm, and Abs″$_{KCN}$ and Abs″$_{aA}$ are obtained in the presence of KCN and antimycin A corresponding to when cytochrome c and cytochrome c oxidase are fully reduced or oxidized.

## Assays for lactate, pyruvate, and insulin

Fractions collected during the experiments were subsequently assayed for lactate, pyruvate, or insulin. Insulin was measured by RIA, and lactate and pyruvate were measured using colorimetric assays using kits per manufacturer's instructions (insulin, Cat no. RI-13K, Millipore Sigma, Burlington, MA; lactate, Cat no. A22189, Invitrogen, Carlsbad, CA; pyruvate, Cat no. MAK332, Sigma-Aldrich). Amounts of lactate, pyruvate, and insulin in inflow samples were insignificant, so rates of production were calculated as the concentration in the outflow times the flow rate and normalized by the amount of tissue.

## Imaging and quantification of cytosolic Ca$^{2+}$

Cytosolic Ca$^{2+}$ was measured by fluorescence imaging of islets after loading them with Fura-2 AM (Invitrogen) as previously described (*Jung et al., 2009*). The perifusion system described above was used to supply buffer with the specified gas composition to a temperature-controlled, 250 μl perifusion dish (Bioptechs, Butler, PA) that was mounted on to the stage of a Nikon Eclipse TE-200 inverted microscope. Results are displayed as the ratio of the fluorescent intensities during excitation at two wavelengths (F340/F380).

## Statistical analysis

When the message to be conveyed by the graph was an illustration of the high resolution and low noise of the data that were generated by the method, then single experiments were shown as indicated, for instance for OCR in response to hypoxia. In most instances, to demonstrate the reproducibility of the data, technical replicates were conducted and the data averaged – that is multiple perifusion channels were run in parallel with pooled tissue or cells batches from multiple animals or flasks of cells. When the goal was to test and show a biological effect, multiple runs were done on different days (for instance comparison of retina and islet recovery of OCR following hypoxia) and statistical significance was determined using Student's *t*-tests carried out with Microsoft Excel (Redmond, WA). With either technical or biological replicates, error bars on time courses were calculated as the average ± the standard error (calculated as SD/$n^{1/2}$). Raw data for all experiments are compiled into Excel spreadsheets saved as a source file that is named with the same descriptor as the figure.

## Acknowledgements

This research was funded by grants from the National Institutes of Health (DK17047) and the National Science Foundation (STTR Phase 2, 1853066). Special thanks to VICI Metronics for providing custom-made permeation tubes. Also, thanks to Gamal Khalil for discussions on interpretation of the lifetime measurements of H$_2$S.

# Additional information

## Competing interests

John Kelly: through his employment at VICI Metronic, has a competing interest for the permeation tubes used in the study. The other authors declare that no competing interests exist.

## Funding

| Funder | Grant reference number | Author |
|---|---|---|
| National Science Foundation | 1853066 | Brian M Robbings<br>James B Hurley |
| National Institute of Diabetes and Digestive and Kidney Diseases | DK17047 | Ian R Sweet |
| National Eye Institute | EY006641 | James B Hurley<br>Ian R Sweet |

The funders had no role in study design, data collection, and interpretation, or the decision to submit the work for publication.

## Author contributions

Varun Kamat, Brian M Robbings, Seung-Ryoung Jung, Data curation, Writing – review and editing; John Kelly, Conceptualization, Methodology, Resources, Writing – review and editing; James B Hurley, Conceptualization, Methodology, Writing – review and editing; Kenneth P Bube, Conceptualization, Formal analysis, Methodology, Software, Writing – review and editing; Ian R Sweet, Conceptualization, Data curation, Formal analysis, Funding acquisition, Investigation, Methodology, Resources, Supervision, Writing - original draft, Writing – review and editing

## Author ORCIDs

James B Hurley ![ORCID] http://orcid.org/0000-0002-7754-0705
Ian R Sweet ![ORCID] http://orcid.org/0000-0002-7565-1663

## Ethics

This study was performed in strict accordance with the recommendations in the Guide for the Care and Use of Laboratory Animals of the National Institutes of Health. All of the animals were handled according to approved institutional animal care and use committee (IACUC) protocols (#4091-01) of the University of Washington. All surgery was performed under sodium pentobarbital anesthesia, and every effort was made to minimize suffering.

## Decision letter and Author response

Decision letter https://doi.org/10.7554/eLife.66716.sa1
Author response https://doi.org/10.7554/eLife.66716.sa2

---

# Additional files

## Supplementary files

• Appendix 1—figure 1—source data 1. Effect of NaHS on insulin secretion rate and cytosolic calcium.

• Appendix 1—figure 2—source data 1. Measurement of depletion of dissolved hydrogen sulfide generated from NaHS.

• Appendix 1—figure 3—source data 1. Effect of hydrogen sulfide on metabolism in liver.

• Appendix 1—figure 4—source data 1. Effect of hydrogen sulfide on electron transport in liver.

• Transparent reporting form

## Data availability

Data for all graphs are contained in Excel files named as the same name as the Figure followed by Source Data.

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

# Appendix 1

## Appendix methods

### Chemicals

Antimycin A, glucose, potassium cyanide (KCN), and succinate were purchased from Sigma-Aldrich.

### Preparation of liver slices

Liver pieces from rats were prepared as previously described (*Rountree et al., 2016*). Ten pieces (about 0.25 × 1 mm [mass = 2.5–3.5 mg per piece]) were loaded into each tissue perifusion chamber without succinate, or four pieces for chambers with succinate for each analysis. The chosen size of the tissue slices was a tradeoff between avoiding hypoxia and retaining a functional unit unperturbed by the trauma of cut tissue. Although there are reports that preparation of liver slices can result in permeabilization of the cells (*Kuznetsov et al., 2002*), experiments comparing uptake and $^{14}$C-sucrose and $^{3}$H-$H_2O$ did not reveal significant membrane permeability in liver slices or retina either before or after perifusion (data not shown). At the end of each experiment, liver samples were weighed, and $O_2$ consumption rate (OCR) measurements were normalized to this mass. Liver pieces had no Cytodex bead layering, but otherwise had the same loading procedure as described for retina chambers, the same supplemented KRB was continuously flowed at a rate of about 95 μl/min after initially loading the tissue at 30 μl/min. Procedures were approved by the University of Washington Institutional Animal Care and Use Committee.

### Measurement of $H_2S$ in solution and headspace

$H_2S$ was measured indirectly by its effect on the emission of a platinum porphyrin (platinum tetrapentafluorophenyl porphyrin) dissolved and baked into a polycarbonate-silicone copolymer (*Sweet et al., 2002*). $O_2$ shortens the lifetime of the emission energy, and $H_2S$ competes with $O_2$'s effect. Therefore, $H_2S$ increases the lifetime, and at constant $O_2$ (and temperature), the lifetime can be used as a measure of changes in both gaseous and dissolved $H_2S$. The platinum porphyrin was affixed to the walls and bottom of 125 ml clear glass bottles, and the lifetime was measured with optical fibers as previously described (*Gilbert et al., 2008*).

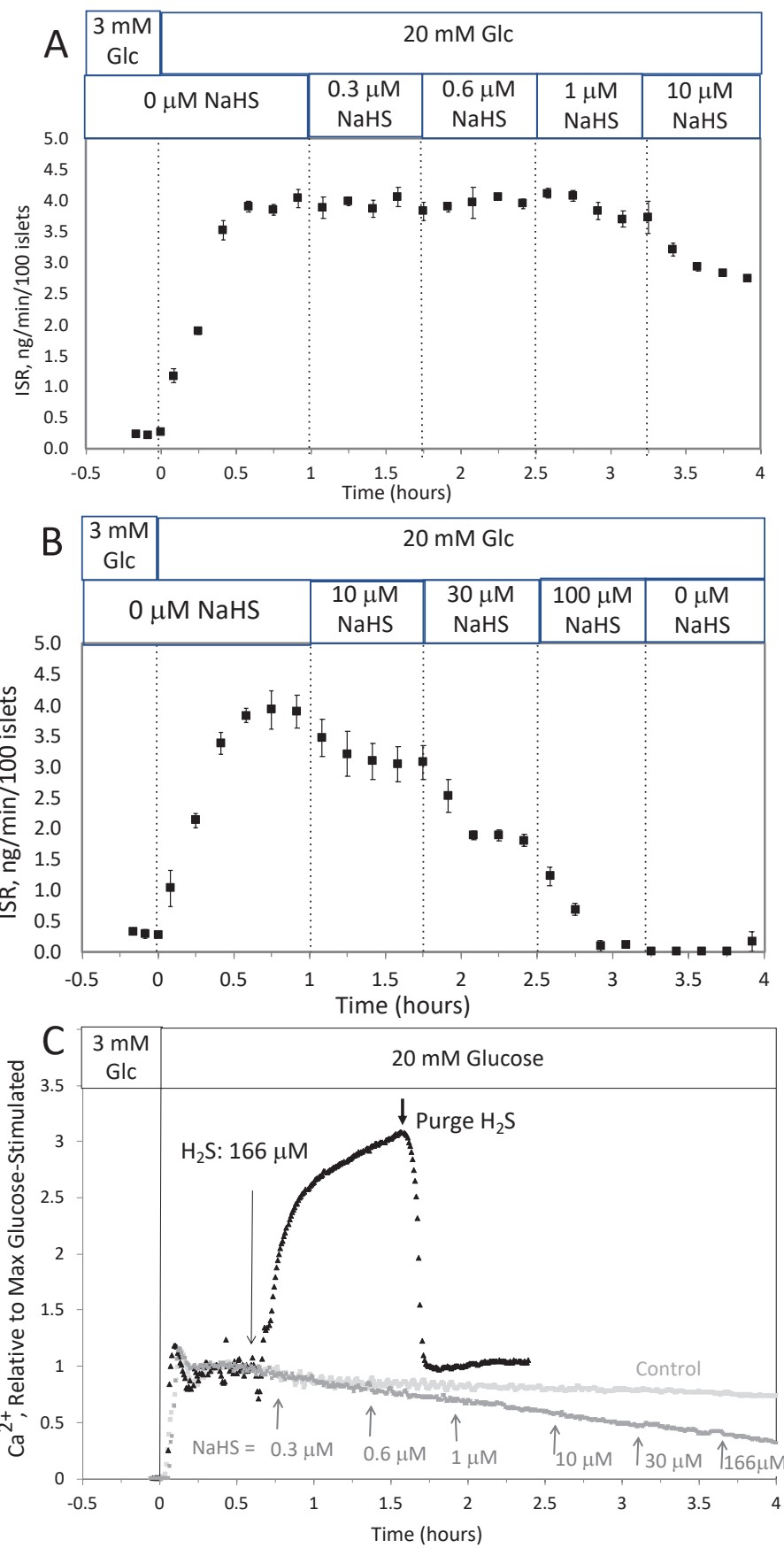

**Appendix 1—figure 1.** Effect of NaHS on insulin secretion rate (ISR) and cytosolic Ca$^{2+}$ (A, B). Rat islets (50/channel) were perifused (flow rate = 200 μl/min), and ISR was measured in response to glucose and exposure to incrementally increasing concentrations of aqueous NaHS as indicated. Data are average ± standard error (SE), $n = 2$. (C) Cytosolic Ca$^{2+}$ in single isolated islets were measured in response to glucose and NaHS. Representative response from four different experiments measuring responses (which were not averaged due to the use of slightly different timings of the protocols). Raw data can be found in a Source Data file named '*Appendix 1—figure 1—source data 1*'.

The online version of this article includes the following source data for appendix 1—figure 1:

- **Appendix 1—figure 1—source data 1.** Effect of NaHS on insulin secretion rate and cytosolic calcium.

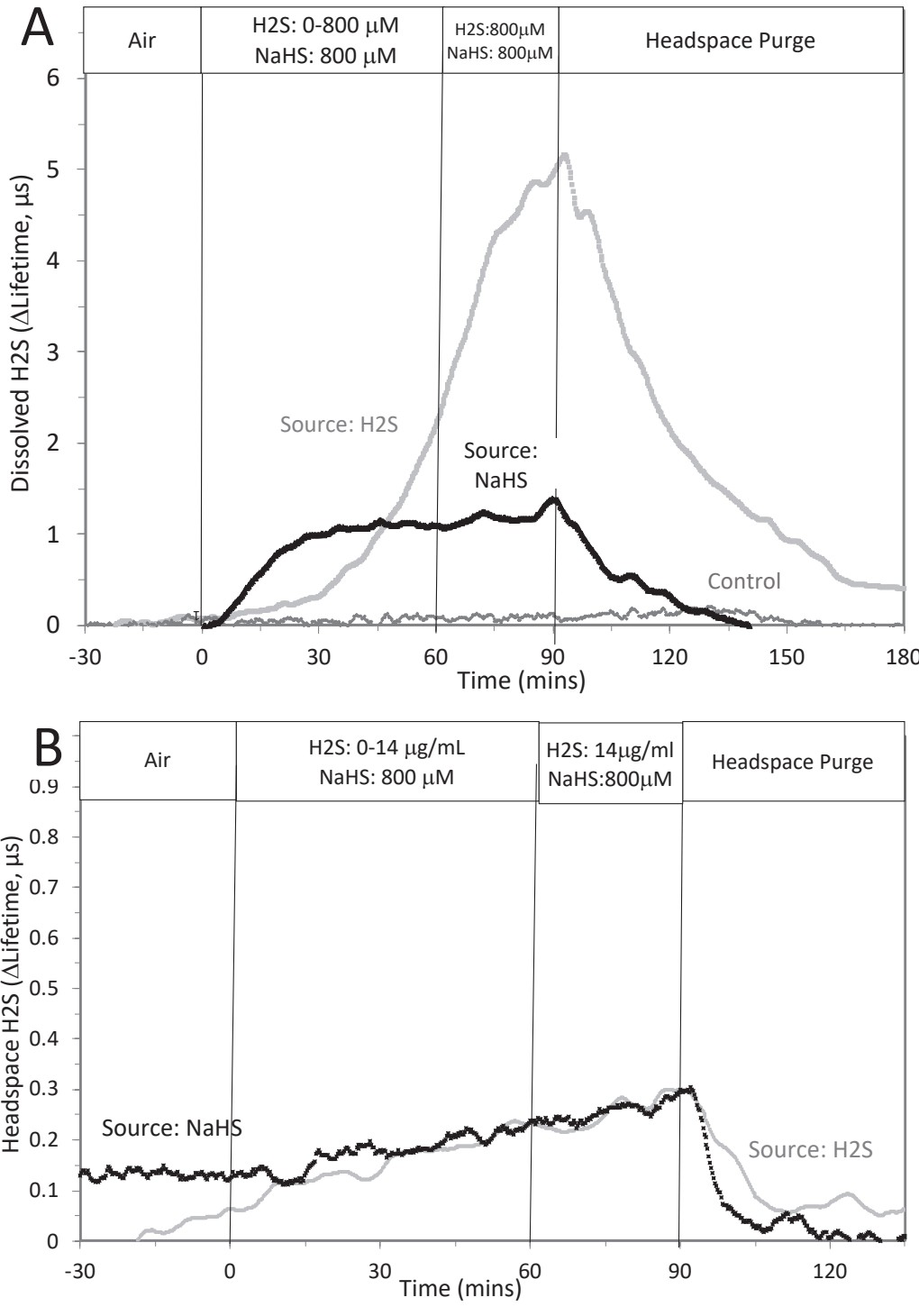

**Appendix 1—figure 2.** Measurement or depletion of dissolved $H_2S$ generated from NaHS. $H_2S$ was measured with sensors placed in KRB (10 ml) and the headspace of a 125 ml sealed bottle in response to either injection of 8 µM NaHS into the solution or attachment of an $H_2S$-emitting permeation tube to the inlet port of the top of the bottle (rate of $H_2S$ permeation = 30 µg/min). Graphs represent lifetime emission reflecting $H_2S$ in the solution (**A**) or the headspace (**B**) from either $H_2S$, NaHS, or no $H_2S$ (control). Legends indicate the source of the $H_2S$. The concentration in solution of $H_2S$ from

*Appendix 1—figure 2 continued on next page*

*Appendix 1—figure 2 continued*

800 μM NaHS, if equilibrium between $H_2S$ and HS was reached in the solution at pH 7.2 is 300 μM (calculated from $[H_2S]$ /($[H_2S]$ +[HS]) = 0.37). The concentration in solution in equilibrium with the headspace after permeation with $H_2S$ gas = 788 μM (calculated from *Equation 1* in the main text ($[H_2S_{aq}]$ = 14,000 ng/ml × 55.5/1000)). The maximal amount of $H_2S$ that could be in the headspace from NaHS in solution was calculated to be 2.5 μg/ml assuming all of the 800 μM NaHS diffused into the headspace. Raw data can be found in a Source Data file named ' *Appendix 1—figure 2—source data 1*'.

The online version of this article includes the following source data for appendix 1—figure 2:

• **Appendix 1—figure 2—source data 1.** Measurement of depletion of dissolved hydrogen sulfide generated from NaHS.

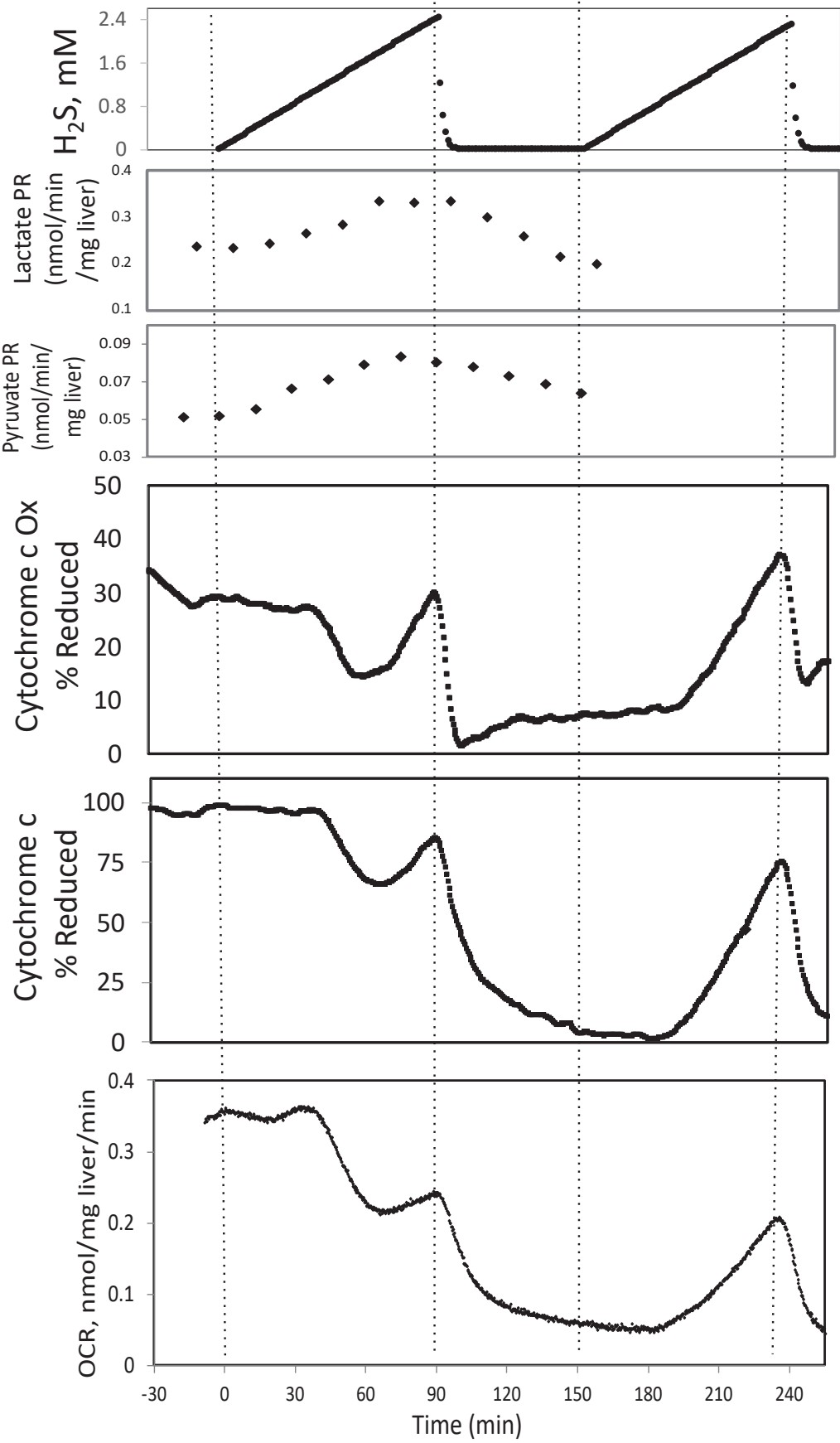

**Appendix 1—figure 3.** Effect of H$_2$S on electron transport chain (ETC) and lactate/pyruvate in rat liver. Liver slices were exposed to increasing levels of H$_2$S until the gas was purged as indicated and an additional ramp of H$_2$S was implemented. O$_2$ consumption rate (OCR), reduced cytochrome c, and cytochrome c oxidase were measured in real time. Fractions were collected for subsequent measurement of lactate and pyruvate. Raw data can be found in a Source Data file named '***Appendix 1—figure 3—source data 1***'.

The online version of this article includes the following source data for appendix 1—figure 3:

- **Appendix 1—figure 3—source data 1.** Effect of hydrogen sulfide on metabolism in liver.

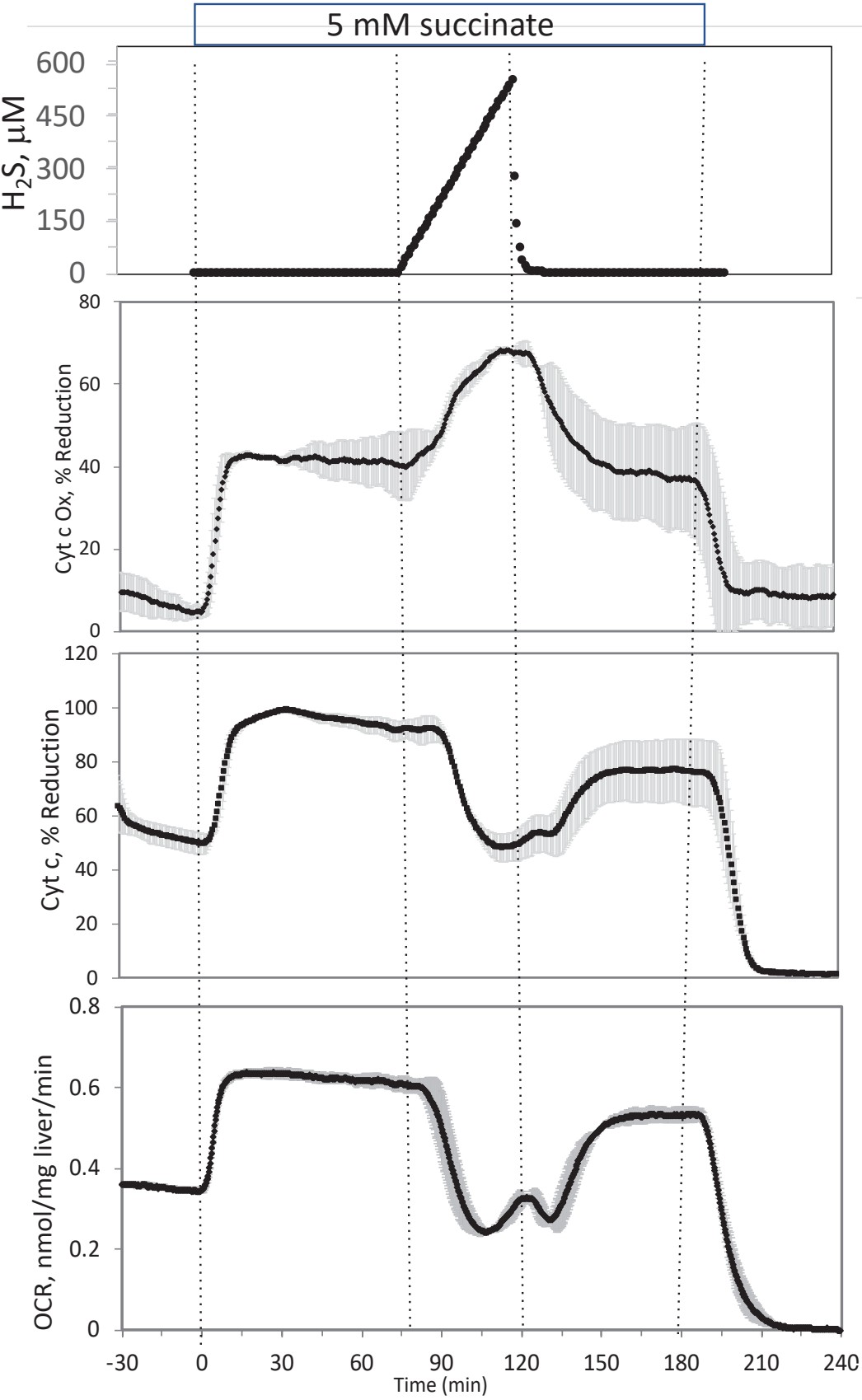

**Appendix 1—figure 4.** Effect of H$_2$S on electron transport chain (ETC) in rat liver. In the presence of 5 mM succinate ($n = 2$), liver slices were exposed to increasing levels of H$_2$S until the gas was purged as indicated. O$_2$ consumption rate (OCR), reduced cytochrome c, and cytochrome c oxidase were measured in real time. Raw data can be found in a Source Data file named '*Appendix 1—figure 4— source data 1*'.

The online version of this article includes the following source data for appendix 1—figure 4:

• **Appendix 1—figure 4—source data 1.** Effect of hydrogen sulfide on electron transport in liver.

## Appendix results

### Effect of NaHS on glucose-stimulated insulin secretion rate by isolated rat islets

To test the ability of NaHS to emulate the of H$_2$S, islets in the flow culture system at incrementally increasing concentrations of NaHS. No effect was seen at concentration below 1 μM (*Appendix 1—figure 1A*), a range where dissolved H$_2$S had both stimulatory and inhibitory effects on insulin secretion rate (ISR). At higher concentrations, NaHS inhibited ISR where the IC$_{50}$ was about 30 μM and each concentration change reached a new steady state within 10–15 min (*Appendix 1—figure 1B*). The effect of NaHS on glucose-stimulated cytosolic Ca$^{2+}$ was measured by incrementally increasing the concentration of dissolved H$_2$S in the inflow (*Appendix 1—figure 1C*). The response did not reach steady state in between increases in H$_2$S, so the concentration dependency was not fully resolved. But in contrast to the brisk increase in cytosolic Ca$^{2+}$ observed in response to H$_2$S, at all concentrations of NaHS tested Ca$^{2+}$ was decreased.

The question arises as to why the response to NaHS did not recapitulate the effects seen by H$_2$S as would be expected if NaHS and H$_2$S immediately equilibrated. However, as has been pointed out, H$_2$S is very volatile and the release of dissolved H$_2$S into the headspace is recognized to be an obstacle to its study (*DeLeon et al., 2012*). We predicted that volatility of H$_2$S explains the difference between effects of H$_2$S and NaHS. To test this, H$_2$S was measured in 10 ml of KRB contained in a sealed 125 ml bottle, while simultaneously measuring the H$_2$S in the headspace above the solution. This was done two ways: with NaHS added to the solution at a concentration of 800 μM (where at pH = 7.2 equilibrium concentration of H$_2$S is 300 μM), and in another sealed bottle when H$_2$S was permeated into the headspace until it reached 14 μg/ml (a concentration that results in dissolved concentration of 788 μM H$_2$S from *Equation 1* in the main text). Using H$_2$S permeation tubes to increase the concentration of H$_2$S in the headspace, the H$_2$S measured (as lifetime of the dye emission) in the solution of the bottle increased, whereas the increase in response to NaHS was barely detectable (*Appendix 1—figure 2A*). Because the solubility of H$_2$S in KRB is about a hundred times that of O$_2$, the sensitivity of the lifetime of the dye emission is very high in solution. The sensitivity of the lifetime to H$_2$S signal in gas was much lower and the time-dependent increase in H$_2$S into the headspace was near the detection limit. Nonetheless, upon purging the bottle, the signals from H$_2$S generated from both the H$_2$S permeation tube and NaHS in solution clearly decreased (*Appendix 1—figure 2B*), confirming transfer of H$_2$S from the NaHS in solution.

### Effect of H$_2$S on liver in the presence and absence of a mitochondrial fuel (succinate)

To test the ability of our system to measure the effects of H$_2$S, rat liver pieces were placed into the perfusion chambers and exposed to steadily increasing concentrations of dissolved H$_2$S. The first effects observed in response to changes in H$_2$S were the increased production rates of lactate and pyruvate (*Appendix 1—figure 3*). About 30 min following the start of the ramp increase in H$_2$S, OCR, reduced cytochrome c, and cytochrome c oxidase all decreased very precipitously for about 20 min, followed by a sudden increase until H$_2$S was purged from the system (*Appendix 1—figure 3*). During the post-H$_2$S phase, all three of these parameters remained low. After a second introduction of H$_2$S, OCR, reduced cytochrome c, and cytochrome c oxidase all increased about 30 min following the start of the ramp. These waveforms suggest multiple points of action by H$_2$S and is consistent with a scenario where H$_2$S can irreversibly inhibit complex one or a step upstream from complex 1 (consistent with decreased cytochromes

and OCR), but also can supply electrons directly to cytochrome c when the level of $H_2S$ reaches higher concentrations.

In the presence of a fuel that enters at complex 2 (succinate), OCR, reduced cytochrome c, and reduced cytochrome c oxidase all increased (*Appendix 1—figure 4*). Under these conditions, $H_2S$ decreased OCR and cytochrome c reduction, but increased cytochrome c oxidase reduction, suggesting that $H_2S$ was inhibiting both at complexes 1 and 4. In contrast to the experiments done in the absence of succinate, post-$H_2S$ metabolism was only slightly different than pre-$H_2S$ energy state reflecting the continued supply of electrons to the ETC when complex one is still inhibited, while complex four inhibition by $H_2S$ seemed to be reversible. Following washout of both $H_2S$ and succinate, reducing power in the mitochondria fell to near 0, as would be expected in the absence of complex one activity.

## Appendix discussion

### Contrasting effect of $H_2S$ and NaHS on ISR and $Ca^{2+}$

The assumption made in most studies of $H_2S$, is that NaHS or $Na_2S$ would rapidly equilibrate with the protonated form of the acid. Whether NaHS or $H_2S$ is added to the solution, the same amount of $H_2S$ would be present in solution after a short equilibration time. Our studies showed that NaHS does not recapitulate the effects of $H_2S$ and that this is caused by rapid depletion of $H_2S$ from the solution via transference to the headspace. The amount of dissolved $H_2S$ in a ml of aqueous solution is about 0.35% of that in equilibria with 1 ml of headspace. This means that unless the volume of the headspace is on the order of 1/100th of the volume of the solution, then the headspace will act as an infinite sink for $H_2S$ in the solution and NaHS will not result in a significant increase in dissolved $H_2S$. This is the case in static or perifusion systems that are commonly used to perform ISR assays. The data support the guidance that when investigating the physiological effects of $H_2S$ in equilibrium with $HS^-$, $H_2S$ should be supplied via the headspace and calls into question the use of $H_2S$ donor molecules where the effects of $HS^-$ but not $H_2S$ will be observed.

Having shown that $H_2S$ in solution generated from NaHS is depleted as a result of transfer into the headspace, we considered two scenarios to interpret the effects of NaHS. Either protonation of $HS^-$ is rapid relative to the diffusion of dissolved $H_2S$ into the headspace and $HS^-$ and $H_2S$ reach equilibrium. Alternatively, diffusion of dissolved $H_2S$ into the headspace is fast, and $HS^-$ generated from NaHS exists in solution in the presence of very low levels of $H_2S$. If the equilibrium between $HS^-$ and $H_2S$ was reached however, then the effect of NaHS would be the same as the effect of $H_2S$, except that the concentration dependency would be right-shifted relative to when $H_2S$ from the headspace is the source of sulfide. However, no concentration of NaHS resulted in an increase of either ISR or $Ca^{2+}$. Thus, it appears that consistent with the low solubility of gases in aqueous solution, and the relatively slow rate of protonation and deprotonation for weak acids, NaHS generates only $HS^-$ when the headspace is similar or larger than the solution, and that $HS^-$ is inhibitory for both ISR and $Ca^{2+}$.

### Control and effects of $H_2S$ on liver

To illustrate its use, we exposed liver to $H_2S$, an established cell signal with multiple effects and mechanisms of action. Both reported mechanisms of action of $H_2S$ in the ETC were observed in this study: inhibition of cytochrome c oxidase (*Khan et al., 1990*) and direct transfer of electrons to cytochrome c (*Vitvitsky et al., 2018*).

As the concentration of dissolved $H_2S$ increased, reduced cytochrome c, reduced cytochrome c oxidase, and OCR all decreased reflecting indicating inhibition of step(s) upstream of complex 1 (which appeared irreversible), and subsequently all three parameters increased consistent with donation of electrons directly from $H_2S$. In the presence of a mitochondrial fuel entering at complex 2 (succinate), $H_2S$ caused a decrease in reduced cytochrome c and OCR, while increasing reduced cytochrome c oxidase consistent with the simultaneous inhibition of complexes 1 and 4. This analysis highlights the benefit of measuring reductive states of cytochromes concomitantly with OCR: interpreting observed changes in OCR in terms of the mechanisms mediating the changes is not possible with OCR measurements alone. The factors mediating the changes in OCR are distinguished by concomitant measurement of both electron pool sizes in the ETC (cytochromes) and flux of electrons (OCR). Overall, the method was able to characterize complex and multiple effects of $H_2S$ on the ETC.

