## [Editor Report]

This paper presents a flow method for measuring the effects of dissolved gases on tissues while having control over tissue concentration. Working with gases can be challenging. The improvements reported here incorporate technology that allows for metabolic characterization of mammalian tissues while precisely controlling the concentration of abundant gases (e.g., oxygen), as well as trace gases (e.g., hydrogen sulfide).

---

## [Decision Letter]

**Decision letter after peer review:**

Thank you for submitting your article "Fluidics System for Resolving Concentration-Dependent Effects of Dissolved Gases on Tissue Metabolism" for consideration by *eLife*. Your article has been reviewed by 2 peer reviewers, and the evaluation has been overseen by Michael Marletta as the Senior and Reviewing Editor. The following individuals involved in review of your submission have agreed to reveal their identity: James N Blaza (Reviewer #1); Kelsey H Fisher-Wellman (Reviewer #2).

Detailed comments are contained within each review. Below I list specific aspects that we conclude are required in your revised manuscript.

1. Controls need to be added to ensure that the tissue is indeed 'intact' and not permeabilized.

2. The reviewers have seen our discussion about NaHS vs. H2S. We would like to see the data your referred to included in the revised paper.

3. Much has been done on hypoxia and the novelty of the tool here would be better appreciated if the focus was on hydrogen sulfide as opposed to oxygen.

*Reviewer #1 (Recommendations for the authors):*

The authors may like to reference the work of Roger Springett in their introduction. His team measured OCR at constant [O2] using a similar, complementary system, although his system was limited to suspension cells. See Kim et al., (2012): Measurement of the mitochondrial membrane potential and pH gradient from the redox poise of the hemes of the bc1 complex and Ripplet et al., (2011): Cytochrome c is rapidly reduced in the cytosol after mitochondrial outer membrane permeabilization.

In Figure 3A and B the authors provide an OCR in terms of nmol/min/10^6 cells, can this be compared to literature values?

A few contractions ('don't' etc) are in the text.

In a few figures u is used rather than μ. In the final, permanent, version it would be good to remedy this.

In Figure 2A and B, the 3 in the 3 mM glucose just becomes '3', which is a little confusing. Could a line label this form without the figure?

*Reviewer #2 (Recommendations for the authors):*

The experiments with H2S are particularly interesting, as this system does seem well suited to investigate the metabolic effects of H2S. Perhaps this line of research could be expanded out in the paper to increase the overall novelty and impact.

The utility of the technique would be more clear if head-to-head comparisons were made with existing technology (Extracellular flux analysis, Oxygraph-2K).

Controls need to be added to ensure that the tissue is indeed 'intact' and not permeabilized.

Can similar experiments be performed in isolated mitochondria? The impact and novelty of the work could be strengthened by detailing the impacts of H2S on mitochondrial bioenergetics across tissues.

In general, there is a lack of mechanistic detail with respect to the observations made.

Throughout the paper, the authors list 'COVID-19' as a potential application. It is not clear how this technology could be used in the context of COVID-19. Could the authors please provide a specific example or remove COVID-19 from the text.

---

## [Author Response]

We have carried out experiments evaluating the permeability of islets, retina and liver before and after perifusion. As shown below in the response to Review #2, the permeabilization was small for all three models and it did not increase during the perifusion experiments.

2. The reviewers have seen our discussion about NaHS vs. H2S. We would like to see the data your referred to included in the revised paper.

We have presented a method that accurately measures the response of insulin secretion and intracellular calcium to dissolved H_2_S. Since this data differs from results that are obtained when using aqueous NaHS with standard methods, we agree that it makes sense to demonstrate why there is a difference between the protonated and deprotonated form of H_2_S. To do this we developed a method to measure dissolved and gaseous H_2_S and used the method to assess the exchange of H_2_S between the two phases. Our results indicate that dissolved H_2_S quickly depletes when the gas headspace in contact with the media does not contain equilibrium levels of gaseous H_2_S. This data has been added as Figure 2A and B in the appendix. Thus, by supplying the H_2_S in the headspace surrounding the inflow media, the effects of an equilibrium ratio of both dissolved H_2_S and HS^-^ are observed. Dissolved H_2_S produced by aqueous NaHS rapidly diffuses into headspaces not supplied with H_2_S and is therefore not a reliable method for investigating the effects of H_2_S. Kenneth Olson previously published an article demonstrating the problems associated with volatility of H_2_S (DeLeon, E.R., Stoy, G.F. and Olson, K.R. Passive loss of hydrogen sulfide in biological experiments. Anal Biochem 421, 203-207 (2012)). We have now appropriately cited this article in the paper. Interestingly H_2_S was observed to be stimulatory for ISR and ca^2+^, where HS^-^ appears to be inert at low levels, and inhibitory at higher levels.

3. Much has been done on hypoxia and the novelty of the tool here would be better appreciated if the focus was on hydrogen sulfide as opposed to oxygen.

We agree that the ability to measure the effect of H_2_S is novel and accurate characterization of the effects of H_2_S and other trace gases will transform the understanding of their role in physiology and pathophysiology. We have now added additional data showing the stimulatory effects of H_2_S on intracellular Ca^2+^ in islets to further demonstrate the utility of the method (Figure 5D) as well as data demonstrating the inhibitory effects of NaHS (Figure 1C in the APPENDIX). As with insulin secretion, H_2_S had the opposite effect on calcium as was observed with NaHS, and resulted in dramatic increases in ca^2+^. This data further supports the conclusion that H_2_S does not operate via the accepted mechanism of opening KATP channels that was established using NaHS. The data showing the transfer of H_2_S from solution to headspace (Figure 2 in the APPENDIX) also confirms the interpretation of the distinct responses to H_2_S vs NaHS.

Nonetheless, we feel that the utility of the methodology to the study of O_2_ should not be de-emphasized. The role of O_2_ in cell and tissue function and pathology is ubiquitous and far reaching, and the many investigators studying the effects of O_2_ will greatly benefit from these methods. It also made sense to us to introduce the method of controlling gas composition first by demonstrating the functionality of our gas equilibration system for controlling abundant gases, followed by the use of permeation tubes incorporated into the gas equilibration system to control trace gases. In summary, we further developed the use of system for the study of a trace gas H_2_S but we did not remove the experiments involving measurement of O_2_ consumption and hypoxia.

Reviewer #1 (Recommendations for the authors):The authors may like to reference the work of Roger Springett in their introduction. His team measured OCR at constant [O2] using a similar, complementary system, although his system was limited to suspension cells. See Kim et al., (2012): Measurement of the mitochondrial membrane potential and pH gradient from the redox poise of the hemes of the bc1 complex and Ripplet et al., (2011): Cytochrome c is rapidly reduced in the cytosol after mitochondrial outer membrane permeabilization.

These are pertinent references to cite, which I have done.

In Figure 3A and B the authors provide an OCR in terms of nmol/min/10^6 cells, can this be compared to literature values?

That is worth doing, so I added a previous paper where we obtained similar results, and I also cited a paper that measured OCR with a Seahorse. Additionally, I have included a comparison to islet OCR. There are about 2000 cells/islets, and glucose stimulated OCR is about 0.8 nmol/min/100 islets or 4 nmol/min/10^6 islet cells. OCR by INS^-1^ cells is about 1 nmol/min/10^6 INS^-1^ cells about ¼ of the rate in islets. This is partially explained by the leakage of lactate out of the cell and also decreased rate of biosynthesis and secretion of insulin.

A few contractions ('don't' etc) are in the text.

Isn’t and Don’t have been removed.

In a few figures u is used rather than μ. In the final, permanent, version it would be good to remedy this.

Agreed. Done.

In Figure 2A and B, the 3 in the 3 mM glucose just becomes '3', which is a little confusing. Could a line label this form without the figure?

True and thanks. We expanded the boxes that contained the text to enable of fitting the units as well as the number.

Reviewer #2 (Recommendations for the authors):The experiments with H2S are particularly interesting, as this system does seem well suited to investigate the metabolic effects of H2S. Perhaps this line of research could be expanded out in the paper to increase the overall novelty and impact.

Although not required for a Tools and Resources Paper, we nonetheless agree that the paper is improved with more novel H_2_S data. We have further developed the investigation of the effects of H_2_S on insulin secretion by carrying out measurements of intracellular Ca^2+^ in islets. Since H_2_S caused an increase in insulin secretion, in contrast to NaHS which inhibited insulin secretion and decreased intracellular ca^2+^, we predicted that H_2_S would increase intracellular ca^2+^. This turned out to be the case, and we have added data showing an increase in intracellular ca^2+^in response to H_2_S as Figure 5D in the paper. In addition, we have added data showing a decrease in ca^2+^in response to NaHS as Figure 1C in the appendix. By presenting both insulin secretion and ca^2+^, we provide more complete evidence that the currently accepted mechanism that the inhibitory effect of H_2_S on insulin secretion involves opening of KATP channels and inhibition of Ca^2+^uptake is incorrect. That mechanism was proposed based on experiments that used NaHS instead of H_2_S. In addition, adding data demonstrating that administering NaHS only delivers HS^-^ without significant H_2_S allows the effects of H_2_S to be assigned to H_2_S vs. HS^-^

The utility of the technique would be more clear if head-to-head comparisons were made with existing technology (Extracellular flux analysis, Oxygraph-2K).

See the response to the Public Review.

Controls need to be added to ensure that the tissue is indeed 'intact' and not permeabilized.

See the response to the Public Review.

Can similar experiments be performed in isolated mitochondria? The impact and novelty of the work could be strengthened by detailing the impacts of H2S on mitochondrial bioenergetics across tissues.

This is a great suggestion, and there certainly is a demand for making the measurements in isolated mitochondria. However, we do not know how to keep the mitochondria immobilized within the flow system so that they would not flow out of the system. We will in the future develop this, but unfortunately cannot include measurements of OCR on isolated mitochondria in the revision.

In general, there is a lack of mechanistic detail with respect to the observations made.

We agree that the results cry out for further investigation of the mechanisms mediating these novel findings. However, we understand that the stated guidelines for an *eLife* Tools and Resources Paper does not demand data, only to present a new methodology positioned to have an impact on a field. This is one of the reasons we selected *eLife* under this category (“Tools and Resources articles do not have to report major new biological insights or mechanisms…”) The examples presented were included to establish the potential impact. Investigation and detailed elucidation of the mechanisms responsible for the regulation revealed by our new instrumentation is beyond the scope of this category of article. Nonetheless the addition of measurements of intracellular *ca^2+^* (a critical factor in the regulation of insulin secretion) is an effective demonstration of how the system can be used to investigate mechanisms mediating the effect of H_s_S.

Throughout the paper, the authors list 'COVID-19' as a potential application. It is not clear how this technology could be used in the context of COVID-19. Could the authors please provide a specific example or remove COVID-19 from the text.

We removed reference to application to COVID.